# Immune Checkpoint Inhibitor-Associated Cutaneous Adverse Events: Mechanisms of Occurrence

**DOI:** 10.3390/ijms26010088

**Published:** 2024-12-26

**Authors:** Abdulaziz M. Eshaq, Thomas W. Flanagan, Abdulqader A. Ba Abbad, Zain Alabden A. Makarem, Mohammed S. Bokir, Ahmed K. Alasheq, Sara A. Al Asheikh, Abdullah M. Almashhor, Faroq Binyamani, Waleed A. Al-Amoudi, Abdulaziz S. Bawzir, Youssef Haikel, Mossad Megahed, Mohamed Hassan

**Affiliations:** 1Department of Epidemiology and Biostatstics, Milken Institute School of Public Health, George Washington University Washington, Washington, DC 20052, USA; eshaq@gwu.edu; 2Research Laboratory of Surgery-Oncology, Department of Surgery, Tulane University School of Medicine, New Orleans, LA 70112, USA; 3Department of Pharmacology and Experimental Therapeutics, LSU Health Sciences Center, New Orleans, LA 70112, USA; tflan1@lsuhsc.edu; 4College of Medicine, Alfaisal University, Riyadh 11533, Saudi Arabia; abdulqaderbaabbad@gmail.com (A.A.B.A.); zainalimkarm33@gmail.com (Z.A.A.M.); msfb742@gmail.com (M.S.B.); ahmed2015khaled2015@gmail.com (A.K.A.); almashoor3@gmail.com (A.M.A.); faroq772928314@gmail.com (F.B.); walamoudi@alfaisal.edu (W.A.A.-A.); 5Department of Radiology, King Saud Medical City, Riyadh 11533, Saudi Arabia; a.bawazir@ksmc.med.sa; 6Institut National de la Santé et de la Recherche Médicale, University of Strasbourg, 67000 Strasbourg, France; youssef.haikel@unistra.fr; 7Department of Operative Dentistry and Endodontics, Dental Faculty, University of Strasbourg, 67000 Strasbourg, France; 8Pôle de Médecine et Chirurgie Bucco-Dentaire, Hôpital Civil, Hôpitaux Universitaire de Strasbourg, 67000 Strasbourg, France; 9Clinic of Dermatology, University Hospital of Aachen, 52074 Aachen, Germany; mmegahed@ukaachen.de

**Keywords:** ICIs, PD-1, CTLA-4, CAR T cells

## Abstract

Immunotherapy, particularly that based on blocking checkpoint proteins in many tumors, including melanoma, Merkel cell carcinoma, non-small cell lung cancer (NSCLC), triple-negative breast (TNB cancer), renal cancer, and gastrointestinal and endometrial neoplasms, is a therapeutic alternative to chemotherapy. Immune checkpoint inhibitor (ICI)-based therapies have the potential to target different pathways leading to the destruction of cancer cells. Although ICIs are an effective treatment strategy for patients with highly immune-infiltrated cancers, the development of different adverse effects including cutaneous adverse effects during and after the treatment with ICIs is common. ICI-associated cutaneous adverse effects include mostly inflammatory and bullous dermatoses, as well as severe cutaneous side reactions such as rash or inflammatory dermatitis encompassing erythema multiforme; lichenoid, eczematous, psoriasiform, and morbilliform lesions; and palmoplantar erythrodysesthesia. The development of immunotherapy-related adverse effects is a consequence of ICIs’ unique molecular action that is mainly mediated by the activation of cytotoxic CD4^+^/CD8^+^ T cells. ICI-associated cutaneous disorders are the most prevalent effects induced in response to anti-programmed cell death 1 (PD-1), anti-cytotoxic T-lymphocyte-associated antigen-4 (CTLA-4), and anti-programmed cell death ligand 1 (PD-L1) agents. Herein, we will elucidate the mechanisms regulating the occurrence of cutaneous adverse effects following treatment with ICIs.

## 1. Introduction

Immunotherapy has become an increasingly popular treatment option for patients suffering from various malignancies. This therapeutic approach is based on the principle of enabling the endogenous immune system to recognize and kill cancer cells [1,2]. Available immune therapeutics include chimeric antigen receptor (CAR) T cell therapy, immune check point inhibitors (ICIs), and cancer vaccines [2,3]. CAR T cell therapies, while not as widely applied as ICIs, have been established as a treatment option for advanced leukemia and lymphomas [4]. ICI-based immunotherapy is an effective treatment strategy for cancers that are heavily infiltrated by immune cells, such as non-small cell lung cancer (NSLCC), as well as cancers with a higher mutational tumor burden (MTB), skin cancer, microsatellite instability (MSI) neoplasms, and virus-related neoplasms [2,3]. While ICIs possess a favorable risk–benefit ratio for their specific safety profile, their molecular action is quite different from those of other systemic therapies [5,6,7,8,9,10,11,12], The most observed ICI-induced adverse effects are mediated by the activation of autoreactive T cells, leading to the occurrence of various adverse effects such as autoimmune diseases as well as gastrointestinal, endocrine, and dermatologic toxicity [5,6,7,8,9,10,11,12]. ICI treatment is associated with dermatological [13,14], gastrointestinal [15,16], pulmonary [7,17], renal [18,19,20], ophthalmologic [21,22], rheumatic [23,24], cardiovascular [25], and hematologic [26] adverse events with varying frequencies and severities. 

These dermatological side effects can have a significant impact on the well-being and quality of life of cancer patients and, in severe cases, may lead to dose modification and interruption or termination of cancer treatment [27]. ICI-induced cutaneous adverse effects are mediated by both immune-dependent and non-immune-dependent mechanism(s). The immune-dependent mechanisms are mediated by factors such as human leukocyte antigen (HLA) alleles and genetic polymorphisms, which are variable among populations besides being drug- and phenotype-dependent [27]. Conversely, non-immune-dependent mechanisms are mediated by factors including abnormalities in genes that encode drug metabolism enzymes, differences in disease type, and drug-related reactions [28].

Although the development of cutaneous adverse effects is common to ICI-based therapy, their occurrence has been shown to be time- and drug type-dependent. Pruritus is one of the most common cutaneous side effects observed in cancer patients treated with either programmed cell death-1 (PD-1)/programmed cell death ligand 1 (PD-L1) or T-lymphocyte-associated antigen-4 (CTLA-4) inhibitors [25,29,30]. The clinical utilization of humanized monoclonal ICIs of PD-1 (pembrolizumab and nivolumab) and PD-L1 (atezolizumab, avelumab, and durvalumab) is widely established as effective cancer therapy [31,32].

The signaling mechanism of anti-PD-1/PD-L1 involves triggering the activation of cytotoxic CD4^+^/CD8^+^ T cells and subsequent killing of cancer cells, causing specific immunological side effects specific to the inhibitors of both PD-1 and PD-L1 [33,34]. These medications can cause numerous skin reactions, which are considered the most common immune-related adverse reactions. Most cutaneous adverse effects range from nonspecific rashes to recognizable skin lesions that sometimes resolve spontaneously and remain within an acceptable skin toxicity profile. However, some may cause life-threatening complications [35,36].

Blockade of the PD-1/PD-L1 pathway has been observed to increase B-cell activation, proliferation, and subsequently production of disease-specific autoantibodies including anti-bullous pemphigoid 180 (BP180) antibody, leading to development of bullous pemphigoid [37]. The development of vitiligo has been reported in melanoma patients after treatment with PD-1 and PD-L1 inhibitors. The occurrence of lichenoid reactions, pruritus, and measles-like eruptions was observed to a low degree, such that immediate discontinuation of therapy was not urgent [38,39].

Although the development of autoimmune bullous diseases in cancer patients treated with ICIs is less likely to occur, the risk of the occurrence of bullous eruption in cancer patients treated with either PD-1 or PD-L1 inhibitors was higher than noted in cancer patients treated with CTLA-4 inhibitor [33,34,35,36]. Meanwhile, the combination of anti-PD-1 (nivolumab) and anti-CTLA-4 (ipilimumab) agents was associated with an increased occurrence of pruritus in cancer patients [37,38]. Notably, the incidence of observed severe cases was low in cancer patients treated with a PD-L1 inhibitor [39,40]. This review largely focuses on ICI-dependent cutaneous adverse effects and the mechanisms of their occurrence.

## 2. Pathophysiology of Immunotherapy-Associated Cutaneous Adverse Effects

Under healthy physiological conditions, the immune system recognizes tumor cells as foreign to the body and triggers the appropriate inflammatory reaction. Figure 1 outlines a proposed model describing the role of both CTLA-4 and PD-1 in the regulation of T-cell activation.

Immune dysregulation is a maladaptive change in molecular control of immune system through various processes mediated by ICIs, a component in the pathogenesis of autoimmune diseases and cancer-dependent mechanisms. The expression of CTLA-4 in activated T cells and Treg cells has been reported to play a crucial role in the regulation of the therapeutic efficacy of ICIs in the treatment of autoimmune diseases and cancer. The adaptive immune system is not the only mediator that is involved in the regulation of anti-tumor immunity [41,42,43]. Many innate leukocytes can distinguish the normal cells from tumor cells in addition to exert tumor-suppressive functions [44,45]. While conventional T cells recognize cancer cells using a rearranged antigen receptor with countless specificities for tumor antigens, innate cells including phagocytic cells, epithelial and endothelial cells, NK cells, innate lymphoid cells, and platelets are characterized by the expression of a fixed set of germline-encoded receptors that are known as pattern recognition receptors (PRRs) [46,47]. These types of receptors play an important role in the regulation of adaptive immune responses [48,49]. Since the enhancement of the adaptive response is based on recognition mechanisms of innate immune cells [50,51]. 

The expression of CTLA-4 in activated T cells and Treg cells has been reported to play a crucial role in the regulation of the therapeutic efficacy of ICIs in autoimmune diseases and cancer [41,52]. CTLA-4 signaling pathway is involved in the maturation of T cells in the thymus as well as in the downregulation of T-cell activation, while the PD-1 signaling pathway is essential for peripheral tolerance to self-reactive T cells [52,53,54].

ICI-induced cutaneous adverse effects result from significant dysregulation of several immunologic pathways in response to CTLA-4 and PD-1 inhibition [55]. Cross-reactivity of antigens on target tumor cells is the primary mechanism that enables immunotherapy to target tumor cells without interacting with self-antigens located on normal host tissues [56]. Unfortunately, immunotherapy treatment does negatively affect normal tissues. For example, cross-reactivity between melanoma-associated antigens on melanocytes treated with ICIs enhances the development of vitiligo during the treatment course [57,58]. T cells that recognize tumor antigens are reactive against skin epitopes [59,60], and can trigger signaling pathways, which lead to skin diseases, such as inflammatory dermatoses [61,62,63]. Inhibition of the PD1/programmed cell death ligand 1 (PD-L1) signaling pathway is associated with increased activation and proliferation of B cells, which leads to the production of inflammatory cytokines, increases in NK cell activity, and antibody production, enhancing autoimmune reactivation [37,64]. In summary, in addition to being effective drugs, ICIs carry the risk of multiple and potentially serious immune-related adverse effects in multiple organ systems. ICI-associated skin reactions include rashes, dry skin, blisters, itching, and vitiligo.

## 3. The Mechanistic Role of Immune Checkpoints in Cancer and Normal Tissues

Immune system regulation is mediated by highly conserved, complex, and tightly regulated mechanisms in both normal and tumor tissues [42,65]. While antigens expressed on tumor cells or on cells suffering from other pathological diseases are recognized by the immune system. However, the presentation of these antigens alone is not sufficient to trigger an effective immune response [49,66]. 

T and B cells are key components of the adaptive immune system. Through their immune properties and their interactions with other immune cells and cytokines in their environment, they form a complex network to achieve immune tolerance and maintain the body’s homeostasis. As is known, the activation of naïve T cells is mediated by two signals. The first one is derived from TCR that recognizes a small part of the antigen to ensure the specificity of the response. Therefore, only T cells that recognize this antigen will be activated. While the second signal that is known as co-stimulatory signal is provided by the co-stimulatory molecule, CD28 [67,68]. Receptors such as CD28 transduce signals that enhance T-cell activation. Conversely, receptors such as CTLA-4 and PD-1 transduce signals, which attenuate T-cell activation [68,69,70]. Figure 2 outlines the molecular mechanisms governing CTLA-4 and PD-1-mediated attenuation of T-cell activation. A balance between “positive” and “negative” signaling is essential in keeping an effective immune response with immune tolerance and reduced autoimmunity [49,71,72]. Negative co-stimulatory molecules, such as CTLA-4 and PD-1/PD-L1, play an essential role in preventing tissue inflammation and autoimmune diseases via attenuation of the excessive activation of T cells [49,73,74]. The regulation of T-cell activation by stimulatory and inhibitory signals coordinates the immune system’s response to many pathophysiological conditions [75,76,77]. However, the quality and the extent of the antigen-specific immune response is determined not only by the quality of positive co-stimulation but also by the loss of positive co-stimulation signals. CD28 and CTLA-4/CD152, two homologous members of the immunoglobulin superfamily, serve as key receptors for T-cell regulation through positive and negative co-stimulation-dependent mechanism [78,79]. Targeting these receptors and their signaling pathways has therefore emerged as a promising therapeutic strategy for modulating the immune response [80,81]. 

The primary function of CD28 Is to facilitate and maintain T-cell response through increased cytokine expression by the primary ligands B7-1 (CD80) and B7-2 (CD86), found on the surface of antigen-presenting cells (APC) [82,83]. Both CTLA-4/CD152 and CD28 are homologous receptors, whose expression is common to both CD4^+^ and CD8^+^ T cells, exerting opposing functions in T-cell activation [84,85]. CD28 is characterized by a stronger affinity interaction with the CD80 dimer and a lower affinity interaction with the CD86 monomer, which mediates T-cell co-stimulation and T-cell receptor (TCR) signaling [77,86]. Blocking the CTLA-4 axis with anti-CTLA-4 antibodies improves the activity of anti-tumor T cells [77,84,85,86,87]. 

Both PD-1 and PD-L1 molecules are involved in the maintenance of immune homeostasis. The co-inhibitory factor PD-1 binds to its ligands PD-L1 or PD-L2 to transmit inhibitory signals in T cells and anti-apoptotic signals in tumor cells [21,86]. Binding of PD-1 is essential in reducing immune cells attack on the body’s own tissue and in maintaining immune balance [59,63,64]. In the human immune system, the binding of PD-1 exerts an immunosuppressive regulatory effect through the Src-homology region 2 domain, through which phosphatase-2 (SHP2) attenuates the T-cell immune response [88,89]. Consequently, targeting immune checkpoints is highly successful in cancer treatment. The most common immune checkpoint receptors include the CTLA-4, PD-1, T-cell immunoglobulin and ITIM domain (TIGIT), T-cell immunoglobulin-3 (TIM-3), and lymphocyte activation gene 3 (LAG-3) [31,83,90,91,92]. The most targeted checkpoints for cancer immunotherapy are CTLA-4 and PD-1 [83,91]. 

The most widely reported mechanism of CTLA-4-mediated inhibition of T cells is attributed to its competitive binding of CD80/CD86 proteins to which it has a greater affinity than CD28 [92,93]. As a result, T cells are deprived of CD28-mediated activating signals. It has also been reported that after binding of CD80/CD86 proteins, CTLA-4 further withholds T-cell stimulatory signaling by transendocytosis of CD80/CD86 molecules from the surface of APCs. [94,95], inhibition of TCR signaling [96], and disruption of the central supramolecular cluster within the immunological synapse [97]. Thus, the disruption of CD28 signaling is widely accepted as the major pathway through which CTLA-4 inhibits T cells [98,99]. PD-1 and CTLA-4 are expressed by activated T cells and most studies of these signaling pathways have focused on their influence on effector responses, which form the basis for checkpoint blockade in cancer [100]. In this context, there is evidence that PD-1 and CTLA-4 act in cis and activate SHP2 phosphatases [101,102,103], which antagonize TCR signals [93,97,104], and thus This weakens the response of effector T cells [105]. Furthermore, the ability of the extracellular domain of CTLA-4 to sequester CD80/86 provides an additional trans-mechanism to limit the function of professional antigen-presenting cells (APCs), which is required for optimal effector T cell activities [106]. Additionally, Treg cells are characterized by their expression of PD-1 and CTLA-4 [107]. Although CTLA-4 is a relevant target for enhancing effector responses in some tumor models [103,108], blockade of CTLA-4 leads to enhanced costimulatory signals and hyperproliferation of Treg cells, resulting in increased immune tolerance [108]. The molecular mechanisms of CTLA-4 and PD-1-mediated attenuation of T-cell activations are outlined in Figure 3.

## 4. Key Mechanisms of Immune System

The ability of the immune system to distinguish between self and non-self host tissues is tightly regulated, involving the recognition and binding of a T-cell receptor (TCR) to an antigen and/or the MHC to the surface of an APC [109,110,111]. Accordingly, other factors also determine whether this binding results in T-cell activation [80,111]. The generation and maturation of a T cell begins in the thymus, where immature cells proliferate and create a wide repertoire of TCRs through genetic recombination [112,113]. When the selection process begins, T cells with strong reactivity to self-peptides can be deleted in the thymus via negative selection [114,115]. Of note, only T cells with insufficient MHC binding can be deleted by an apoptotic mechanism, not those that weakly respond to MHC molecules and self-peptides [116,117]. 

While the negative selection of T cells with autoreactivity is biologically efficient, some TCRs may have specificity that is cross-reactive with self-antigens [118,119,120]. Numerous immune checkpoint pathways can regulate the activation of T cells during the immune response, particularly CTLA-4 and PD-1, which perform this task through peripheral tolerance [121,122,123]. Both the CTLA-4 and PD-1 pathways operate at different stages of an immune response. CTLA-4 suppresses potentially autoreactive T cells at the initial stage of naive T-cell activation [78,90], whereas PD-1 regulates activated T cells at later stages of the immune response in peripheral tissues [41,95]. Thus, targeting the immune checkpoint pathways to eliminate tumor cells via T-cell-dependent mechanisms exhibits exciting potential in cancer immunotherapy [123,124,125,126,127,128]. 

### 4.1. CTLA-4 Pathway

As mentioned above, T-cell activation is a complex process requiring stimulatory signaling. While the binding of TCR to MHC is the main cause of T-cell activation. However, the involvement of other co-stimulatory signals is required [64,77,80,81,82,83,84,85,86,87,88,89,90,91,92,93,94,95,96,97,98,99,100,101,102,103,104,105,106,107,108,109,110,111,112,113].CD28 is the best studied co-stimulatory glycoprotein, and its function is crucial for the co-stimulation of naive T lymphocytes [123,128]. CD28 also protects T cells from apoptosis and increases cell proliferation as well as their cytokine secretion [123,128]. Binding of B7-1 (CD80) or B7-2 (CD86) molecules with CD28 receptors on T cells are required to promote signaling within T cells [84,85,87]. Notably, adequate amounts of CD28:B7-1/2 binding is required to trigger T-cell proliferation, survival, and differentiation in response to the production of cytokines such as interleukin-2 (IL-2) [89,91,129]. Although CTLA-4 that is known as CD28 homolog and characterized by its higher binding affinity for B7 [53,130]. However, binding of CTLA-4 to B7 alone is not enough to trigger a stimulatory signal [49,131]. Since the competition between CD28 and CTLA-4 for B7 binding 7 can prevent the co-stimulatory signaling [89,94,132]. As is widely known, binding of CTLA-4 to B7 is inhibitory, and its affinity and avidity is higher than those of CD. Thus, the generation of inhibitory signals is expected to counteract stimulatory signals [42,79,85,95]. We proposed model for the mechanisms regulating the inhibitory signals leading to direct inhibition of the TCR immune synapse. Inhibition of CD28 signaling pathway, or increased T-cell mobility is then essential to reduce the interaction with APCs [91,95,133,134]. Thus, the relative amount of CD28:B7 binding versus CTLA-4: B7 binding determines whether a native T cell become activated or anergic [77,119,133]. Indeed, binding of CTLA-4 to B7 can generate inhibitory signals to counteract CD28:B7 and TCR: MHC-dependent stimulatory signals [95,134,135,136].

In contrast to effector T cells, Tregs constitutively express CTLA-4, while their function of Tregs is to control effector T cell functions and to serve as key player in maintaining peripheral tolerance [42,92]. Figure 4 demonstrates mechanisms of CTLA-4/B7 and CD28/CD80/86 pathway-dependent tumor immune escape.

### 4.2. PD-1 Pathway

The PD-1/PD-L1 signaling pathway is responsible for the induction and maintenance of immune tolerance in the tumor microenvironment. The activity of PD-1 and its ligands PD-L1 or PD-L2 is essential to control the activation, proliferation, and cytotoxic secretion of T cells to suppress abnormal antitumor immune responses [21,40,59]. Like CTLA-4 signaling, PD-1 binding inhibits T-cell proliferation, production of interferon-γ (IFN-γ), tumor necrosis factor-α (TNF-α), and IL-2 leading to reduced T-cell survival [137,138].

Generated PD-1 signals prevent the phosphorylation of TCR signaling intermediates, terminate early TCR signaling, and reduce T-cell activation [90,138]. PD-1 expression is a hallmark of “exhausted” T cells [139,140]. Both CTLA-4 and PD-1 binding have similar negative effects on T-cell activity. However, the downregulation of CTLA-4 and PD-1 receptors-dependent mechanisms by ICIs is different. Since CTLA-4 expression is restricted to T cells, while those of PD-1 expression is more likely to occur on activated T cells, B cells, and myeloid cells [31,64,110,137,141].

Whereas CTLA-4 acts during the priming phase of T-cell activation, PD-1 acts during the effector phase, mostly in peripheral tissues [110,138,142]. The distribution of PD-1 ligands differs from that of CTLA-The expression of B7, the ligand of CTLA-4, is restricted to APCs that are typically located in lymph nodes and in the spleen [90,118,140,142].

PD-L1 and PD-L2 expressions are common in various tissues, such as leukocytes, non-hematopoietic cells, and non-lymphatic tissues [63,143,144]. The expression of PD-L2 in a variety of other immune and non-immune cells, such as monocytes and dendritic cells, is regulated by microenvironment-dependent mechanisms [130,131,132]. Of note, the exhaustion of T cells occurs during chronic infections and cancer and is characterized by T-cell dysfunction leading to the suboptimal control of infections and tumors. 

PD-L1 and PD-L2 signaling play an important role in the activation of T cells [63,143,144]. Thus, inhibition of PD-L2 binding is associated with an increase in T helper 2 (TH2) cell activity [145,146], while binding of PD-L1 to CD80 is associated with the inhibition of T-cell responses [144,147,148].

Although Tregs express both PD-1 and CTLA-4, the function of PD-1 expression on these cells is unclear [149,150]. PD-L1 has been shown to contribute to the conversion of naive CD4^+^ T cells into Treg cells [151,152], by inhibiting T-cell responses and promoting the induction and maintenance of Tregs [151,152].

Consistent with these reports, PD-1 blockade can reverse Treg-mediated suppression of effector T cells in vitro [80,112,148,152]. Binding of PD-1 to its ligands is essential to attenuate the immune response in T cells involved in an effector response. [110,146,152,153]. Figure 5 outlines the mechanisms of PD-1/PD-L1 pathway-dependent tumor immune escape.

## 5. Molecular Mechanisms of ICI-Induced Cutaneous Adverse Events

Immune checkpoint blockade by specific inhibitors, namely, anti-CTLA-4 and anti-PD-1 inhibitors, is essential in blocking the inhibitory signals of T-cell activation, which allows tumor-reactive T cells to release an effective anti-tumor response [90,139]. These regulatory mechanisms are essential in maintaining immune responses under normal physiological conditions and restraining host autoimmunity [125,154,155]. Central tolerance is mediated through clonal deletion of high-affinity self-reactive clones during negative selection in the thymus [154,155]. Immunologic tolerance is mediated by multiple distinct central and peripheral events [156,157,158], whereas peripheral tolerance is mediated through Treg, T-cell anergy, cell-extrinsic tolerogenic signals, and peripheral clonal deletion [115]. During tumor development and progression, the host immune system can exert strong selective anti-tumor activity [139,157,158,159]. Consequently, tumor cells protect themselves by releasing their own factors to initiate immune suppressive and tolerance mechanisms [159,160,161]. Non-specific immunologic activation, such as autoimmune inflammatory disease, are immune-related adverse events [139,158,159]. As stated above, cutaneous adverse effects mediated by ICIs significantly impact patients’ quality of life [160,161,162]. Early detection and investigation of the mechanisms underlying the occurrence of ICI-related cutaneous adverse reactions is crucial for the development of clinically relevant approaches to minimize these adverse reactions during the treatment course. As is established and widely recognized, immune checkpoint molecules serve as negative regulators that maintain immune homeostasis and regulate immune responses to prevent autoimmunity [162,163]. For self-maintenance and progression, tumor cells develop mechanisms to evade the action of the host’s immune system by activating immune checkpoint signaling pathways, which have the potential to suppress immune responses [133]. Blockade of immune checkpoint signaling pathways has been proved effective in restoring suppressed antitumoral cytotoxicity [164,165]. Thus, inhibition of immune checkpoints offers an exciting therapeutic promise for cancer patients.

Although the exact mechanisms responsible for ICI-induced cutaneous side effects are not yet fully understood, several mechanisms have been proposed. These include activation of T cells against common antigens both in tumor and normal cells, enhanced inflammatory cytokines release, production of immune-related antibodies, and induction of HLA variants [164,165]. Stimulation of B cells and humoral immune responses during ICIs treatment increases proinflammatory cytokines production [166,167].

The pathomechanisms through which B cells and humoral immunity contribute to the development of ICI-associated cutaneous adverse reactions are diverse and involve several key aspects of immune regulation [162,166]. Under normal conditions, B cells are tightly controlled and prevented from producing autoantibodies that target host tissue [167,168]. Thus, disruption of this immune checkpoint by specific inhibitors leads to destruction of the corresponding inhibitory pathways, which regulate B cell tolerance [169]. Activation of autoreactive B cells results in the production of autoantibodies directed against self-antigens [92,136]. These autoantibodies form immune complexes involved in the development of several types of autoimmune diseases [170,171], including bullous pemphigoid [172]. 

Autoimmune bullous skin diseases are characterized by autoantibodies and T cells specific to structural proteins maintaining cell–cell and cell–matrix adhesion in the skin [171,173]. These autoantibodies belong to several IgG subclasses exhibiting different functional properties and may determine the pathogenic potential of IgG antibodies [174].

In pemphigus diseases, binding of IgG to keratinocytes causes intraepidermal blisters, with IgG4 autoantibodies mediating acantholysis [64,175,176]. The induction of morbilliform by ICIs occurs through a cytotoxic T-cell-dependent mechanism and is classified as a type IV immune reaction ICI-associated inflammatory skin reactions that is known as a group of several skin rashes including maculopapular, lichenoid, psoriasiform, and eczematous rashes as well as cutaneous adverse events including erythema multiforme; palmoplantar erythrodysesthesia; and neutrophilic dermatoses, such as sweet syndrome [29,177,178,179].

The primary function of the PD-1/PD-L1 pathway is to control the induction and maintenance of immune tolerance within the tumor microenvironment [62,63]. This pathway inhibits T-lymphocyte activation, which reduces cytokine production, and exhausts CD8^+^ T lymphocytes [37,158]. Inhibition of the PD-1/PD-L1 pathway activates cytotoxic T cells that trigger anti-tumor activity and induce morbilliform that is known as a type IV hypersensitivity reaction [37,163,180,181]. Figure 6 demonstrates the mechanisms associated with ICI-induced cutaneous adverse effects. Of note, ICI-induced cutaneous adverse effects are reversible with systematic corticosteroid use. Glucocorticosteroids, unfortunately, are associated with multiple side effects that may impact the antitumor response [182,183]. Table 1 summarizes systemic agents used for the treatment of ICI-induced cutaneous adverse effects.

## 6. Immunotherapy-Associated Cutaneous Adverse Events

ICI-associated cutaneous adverse reactions are characterized by their delayed occurrence and longer duration when compared to the characteristics of the classic chemotherapy-associated adverse events [191,192]. Depending on the type of cutaneous adverse effects, their time of occurrence ranges from a few weeks to several months after the treatment initiation [187,193]. Unfortunately, the relationship between cutaneous adverse effects and the dose or time of ICI exposure is not fully understood. ICIs can trigger a variety of skin reactions, which represent a reactivation or worsening of existing dermatosis or a new development [62,191]. Although the classification of ICI-induced cutaneous adverse effects is still unclear, the most known cutaneous adverse effects derived from clinical observation include rash or inflammatory dermatitis that encompass erythema multiforme, lichenoid, eczematous, psoriasiform, morbilliform, and palmoplantar erythrodysesthesia. 

### 6.1. Eczematous Dermatitis

Eczema is a chronic inflammatory skin condition that causes dry, itchy, red skin. It can also cause scaly patches, a swollen rash, weeping blisters, and scaly flaking. The appearance of the rash can vary depending on skin tone, with individuals with darker skin tones developing purple, brown, or gray rashes, while patients with lighter skin tones develop pink, red, or purple rashes [37,194]. The incidence of developing eczema has been reported to increase at an approximately constant rate after the treatment of ICIs [64,110]. A cumulative incidence rate of 25% was shown to develop first eczema within 10.3 months, a cumulative incidence rate of 33% of patient within 13.8 months and a cumulative incidence rate of 40% within 20 months after the treatment has been initiated [64]. Treatment of cancer patients with CTLA-4-based therapy is therefore associated with eczematous dermatitis occurrence [65,66]. The symptoms associated with eczematous dermatitis are generally pruritic with poorly demarcated patches or papules and erythema, which may merge into larger plaques or patches [66,78]. The rash commonly covers large areas such as the trunk and limbs, sometimes even reaching the face. The treatment for ICI-associated eczematous eruptions involves topical corticosteroids, calcineurin inhibitors, and emollients [195,196,197].

Low-potency corticosteroids are commonly prescribed to treat body areas such as the face, genital area, and axillary and inguinal folds, areas characterized by higher trans-epidermal absorption [190,198]. More severe manifestations on the trunk can be treated by medium to high-potency corticosteroids [199,200]. Although environmental interventions together with topical devices including emollients, corticosteroids, and calcineurin inhibitors represent the mainstay of treatment, systemic treatments are reserved for severe cases [201,202]. Phototherapy represents a valid second-line intervention in those cases where non-pharmacological and topical measures have failed. Different forms of light therapy are available and have showed varying degrees of beneficial effect [203,204]. The most common phototherapeutic options of severe eczematous dermatitis are the natural sunlight, narrowband (NB)-UVB, broadband (BB)-UVB, UVA, UVA1 that can administrate either as monotherapy or in combination with systemic corticosteroids [203,204].

### 6.2. Morbilliform Reactions

A morbilliform, or maculopapular, drug eruption develops via a common drug hypersensitivity reaction. It is characterized by a symmetrical rash with red macules and papules that may appear one to two weeks after drug exposure. Other symptoms include mottling or plaques, fever, and bleachable rash areas [205,206,207]. As the mucous membranes are typically spared, it is possible to distinguish between mucous membranes less and more erupted [205,206]. Identified rashes in immunotherapy treated cancer patients appear either as itchy, bleaching, erythematous patches or papules on most of the body surface, especially on the head, the palms, and the soles of the feet [205,206,207]. The appearance of morbilliform maculopapular rash is mediated by a mechanism similar to those of type IV hypersensitivity [208,209,210,211,212]. In these mechanisms, the cytotoxic T cells act as effector cells [62,213,214,215]. Thus, during immunotherapy, activated cytotoxic T lymphocytes can directly damage target cells through the release of cytotoxic cytokines, including perforin, granulysin, and either granzyme or granzyme B, and through physical interaction via the FasL/FasR signaling pathway [215,216,217].

As is known, morbilliform rashes are very common in patients treated with anti-CTLA4 therapy. Up to 14 to 26% of patients receiving ipilimumab and up to 55% of patients receiving combination anti-CTLA4/PD-1 inhibition therapy were found to develop morbilliform rashes [208,209,210,211,212]. While the lower frequency rates were noted in patients received anti-PD-1 and anti-PD-L1 therapies, up to 20% of treated patients developed morbilliform rashes [209]. While the highest morbilliform reactions were found to occurring in fewer than 2% of patients received monotherapy and fewer than 5% of patients received combination regimens [209]. However, early diagnosis and treatment is necessary to continue oncological therapy at an effective dose. [64,210].

Although the incidence of cutaneous adverse effects ranges from 30 to 60% in patients treated with ICIs [203,204,205,206,207,208,218,219,220,221,222,223], the frequency of cutaneous adverse is variable among patients and ICI-dependent [62]. For example, patients who received anti-CTLA-4 monotherapy demonstrated higher incidence of cutaneous adverse effects (44–59%) when compared with those received anti-PD-1 (34–42%) and anti-PD-L1 (up to 20%) monotherapy [62]. The highest incidence of cutaneous adverse effects (5–72%) was noted in patients treated with the combination of anti-PD-1 and anti-CTLA-4 agents [210,212,213]. The severity analysis of cutaneous adverse effects was observed in about 25% of patients treated with anti-CTLA-4 agents [210,212,213], and only 2.4% of treated patients had grade 3 and 4 severe adverse effects [214]. More important, the incidence of cutaneous adverse effects of grades 3 and 4 were found to be much higher in patients treated with anti-PD-L1 monotherapy (7.2%) compared to those observed in patients treated with anti-PD-1 monotherapy (2.3%) or those in patients treated with anti-CTLA-4 monotherapy (4.7%) [215]. While the highest incidence was noted in patients treated with anti-PD-L1 in combination with anti-CTLA-4 (14.5%), when compared with the incidence of cutaneous adverse effects in patients treated with anti-PD-1 combined with anti-CTLA-4 therapies (5.4%) [215].

The appearance of a morbilliform (maculopapular) rash on the trunk and extremities was noted in tumor patients following treatment with either anti-CTLA-4, PD-1 or PD-L1 therapy [206,207]. A morbilliform rash often starts on the chest and spreads to the arms, legs, and neck; it is characterized by the appearance of flat pink or red spots on the skin after the onset of such rashes. Topical steroids, topical calcineurin inhibitors, systemic antihistamines, and topical emollients are often used to treat ICI-associated morbilliform maculopapular rash [201,202,203,216,217,218]. As mentioned above, low-potency topical steroids are preferred to treat the face, axilla, and groin, with treatment options depending on the severity of the disease [204,205,219,220].

Treatment of morbilliform maculopapular rash depends on the severity of the rash and is based on the Common Terminological Criteria for Adverse Events (CTCAE) [198]. For example, treatment of grade 1 includes symptomatic measures such as application of topical moisturizers and topical potent or high-potency corticosteroids to the affected areas [221,222,223]. In grade 2, oral antihistamines are also administered for treatment [224,225]. Systemic corticosteroids (0.5–1 mg/kg/day of prednisone equivalent), in addition to the symptomatic measures used in grades 1–2, are maintained in grade 3 rashes, but only after excluding other diseases requiring specific treatment (e.g., psoriasis). Oral steroids are tapered after improvement of MIT over 4 weeks [224,226,227].

### 6.3. Lichen Planus 

Lichen planus is a clinically and histopathologically typical, relatively common pruritic papulosquamous skin disease characterized by purple, flattened, polygonal papules that preferentially affect the distal parts of the extremities and the lower back [62,198]. In addition to rashes, it can also affect mucous membranes and nails or cause scarring alopecia [228,229]. In addition to rashes, it can also affect mucous membranes and nails or cause scarring alopecia [228,229]. On the oral and genital mucosa, lichen planus can also form lacy white patches, sometimes with painful sores [160,200,201]. A grid-like network of white lines covering the lesions are most easily seen on the oral mucosa, where erosions occur. Drug-induced lichen planus or lichenoid drug rash is often photolyzed and cannot be distinguished from idiopathic lichen planus [211,215]. Bullous lichen planus is a rare variant of lichen planus in skin and mucous membranes with an unknown cause [230,231,232,233]. Lichen planus pemphigoids are different from lichen planus in that blisters are more frequent and extensive, with a longer disease course [234,235]. Despite the close relationship between ICIs and/or other drug treatment and infection with viral hepatitis, the factors outlying the appearance of lichen planus or bullous lichen planus have not received investigation [236]. ICI-associated lichenoid eruptions may vary in appearance, but most commonly they appear as pruritic, erythematous to purple, flat papules or plaques on the extremities and trunk [58,231]. T cells that recognize tumor antigens are reactive against skin epitopes. In addition to be more frequently than maculopapular rash, the occurrence of anti-PD-1 and PD-L1 therapy-associated lichenoid eruption is delayed [207,216]. Inverse lichenoid eruptions have also been reported, including bullous lichen planus pemphigoids, erosive and hypertrophic variants, and oral lichenoid eruptions [232,233]. Lichenoid and lymphocytic infiltrates have been observed at the tissue border, coinciding with basal vacuolar changes [232,233]. Development of epidermal spongiosis, parakeratosis, eosinophil infiltration, and necrosis have also been reported during ICI treatment [234,235]. ICI-associated lichen planus is treated primarily with topical corticosteroids [60,61]. Other therapeutic options, including oral corticosteroids, phototherapy, acitretin, and cyclosporine, which have efficacy in refractory cases, have been suggested for ICI-lichen planus [201,205].

### 6.4. Psoriasis

Psoriasis is a chronic disease that is mediated by the cells and molecules of both the innate and adaptive immune systems mediated by the overactivation of immune system [200,226,236]. Although psoriasiform dermatitis has similarities to psoriasis, it encompasses a broader spectrum of skin conditions with a psoriasis-like appearance [237]. The development of psoriasis in cancer patients is one of the common cutaneous side effects observed for patients following ICI treatment [238,239]. ICI-induced psoriasis can be divided into two types: one of them is referred to as new-onset psoriasis, and the other one is reactivated psoriasis [62,240]. 

The development of psoriasis in cancer patients is one of the common cutaneous side effects observed for patients following ICI treatment [238,241]. ICI-induced psoriasis can be divided into two types: one of them is referred to as a new-onset psoriasis and the other one is reactivated psoriasis [62,241]. Data from the European Network on Cutaneous Adverse Events of Oncology Medicines showed that the epidemiological analysis of 115 cases of psoriasis derived from cancer patients treated with ICIs revealed that 30% of the patients’ reactivated psoriasis and 70% of the patients exhibited new-onset psoriasis [239,240,241]. Notably, patients included in the abovementioned study 86% of patients received anti-PD-1 therapy while only 14% of the patients received anti-PD-L1therapy [241].

The treatment of cancer patients with anti-PD-1 agents is expected to trigger the activation of both innate and adaptive immune systems. Consequently, the cells that may be affected by PD-1 inhibitors such as dendritic cells, T helper cells, and Treg cells are characterized by their ability to secrete cytokines including interferon-gamma, IL-1, IL-17, and IL-22 [242,243] and thereby play an important role in the pathogenesis of both new-onset psoriasis and reactivated psoriasis [243,244]. 

As is known, neutrophils are a key driver of the innate immune response that plays an important role in the axis of psoriasis pathogenesis [242]. Since the migration of neutrophils from blood vessels into the epidermis to form a Munro or Kogoj microabscess is the typical pathological feature of psoriasis [245,246]. However, the mechanisms whereby, the neutrophils contribute to the occurrence of psoriasis is attributed to their ability to secrete various cytokines and chemokines such as TNF-α, IL-17, and IL-36 family factors, as well as neutrophil extracellular traps [247]. The mechanisms whereby the dendritic cells contribute to the development of psoriasis during the treatment of cancer patients with PD-1 inhibitors is based on the ability of tumor cells and tumor-infiltrating immune cells (e.g., macrophages and myeloid dendritic cells) to express PD-L1 [243,248]. Consequently, PD-1/PD-L1 signaling in tumor microenvironment can mediate the inhibition of T cells [249,250]. Thus, the treatment of cancer patients with PD-1 inhibitors results in the have been demonstrated to directly induce Interferon γ (IFN-γ) production of activated T cells, release of IL-12 by intra-tumor dendritic cells subpopulations [251,252]. As is widely reported, IL-12 is involved in the regulation of anti-tumor T-cell immunity and stimulation of T-cell proliferation that, in turn, can secrete various cytokines to initiate a positive feedback inflammatory loop [253,254]. Additionally, IL-12 has the potential to inhibit eomesodermin, a key regulator of T-cell exhaustion [255]. To that end, IL-12 is suggested to be one of the factors that play an important role in the onset and development of psoriasis [256]. Additionally, the involvement of macrophages in the development of psoriasis is considered. Since the increased macrophages in psoriatic lesions has been reported [257]. Macrophages can also promote psoriatic angiogenesis via mechanisms mediated by the expression of Vascular Endothelial Growth Factor (VEGF), TGF-β, Platelet-Derived Growth Factor (PDGF), and TNF-α [258,259]. Additionally, macrophages have been reported to play a role in positive feedback of inflammatory factors including IL-1 and prokineticin 2 in psoriasis [260]: 

Cancer patients with a history of psoriasis are more likely to develop cutaneous psoriasis after ICI treatment than inexperienced patients with psoriasis [261,262]. ICI-associated psoriasis is related to plaque psoriasis [211,263]. ICI-induced psoriasis includes psoriasis vulgaris, lichenoid features, spongiosis, and eosinophil infiltration [66,248,264]. In many cases lichenoid reactions disappear spontaneously after ICIs discontinuation [203,261]. Psoriasiform eruptions can be resistant to conventional psoriasis treatments and may require more targeted therapies [66,69]. In some cases, medication with TNF-α inhibitors has successfully treated ICI-induced psoriasis [61,66]. Cancer patients with pre-existing psoriasis have shown improvement during ICI therapy, suggesting a complex interaction between the immune system and these medications [212,265,266]. Common therapeutic modalities for psoriasis include topical steroids, topical vitamin D analogs, topical steroids and topical vitamin D analogs as well as narrowband ultraviolet B light phototherapy [266,267]. In addition to skin-directed therapies, systemic treatments include acitretin, methotrexate, and apremilast [268,269].

### 6.5. Bullous Disorders

Bullous diseases are a group of rare autoimmune skin diseases that cause blistering of the skin and/or mucous membranes. The blisters can range from mild to severe and are often filled with fluid [270,271,272,273]. Although the exact cause of bullous disease is unknown, it is believed to be caused by autoantibodies directed against the epidermis or the dermoepidermal junction [238,269].

Unlike many cutaneous adverse effects that occur within a few months of ICIs treatment, the occurrence of Bullous disorder can appear for the first several months after the treatment is initiated [274,275]. Compared to classic bullous disease, bullous disease in patients treated with ICIs has a prolonged pruritic prodromal phase and usually requires treatment with high-dose corticosteroids for effective symptom management [274,275]. Treatment of ICI-associated bullous disease can be controlled via administration of topical or systemic steroids [163,276]. The most common steroid-based therapies used in the treatment of ICI-associated bullous disease include doxycycline, niacinamide, omalizumab, rituximab, dapsone, methotrexate, and plasma exchange [246,247].

Although bullous disorders are not common as an adverse effect among ICI treated cancer patients [277], bullous pemphigoid, the most common blistering disorder, occurs in about 1% of patients treated with PD-1 or PD-L1 inhibitors [34,209,278]. Notably, the occurrence of bullous pemphigoid was observed in cancer patients treated sequentially with PD-1 and CTLA-4, but not in those treated with ipilimumab monotherapy [269,279]. In ICI-induced bullous pemphigoid, a longer period of rash-free pruritus and a longer time interval between symptom onset and diagnosis are observed compared to idiopathic bullous pemphigoid. [276,280].

The induction of bullous disorders by PD-1/PD-L1 inhibitors is thought to be mediated by both T cell and B cell dysregulation-dependent mechanisms. In T cell-independent humoral immunity, PD-1/PD-L1 blockades can enhance B cell activation, releasing the production of disease-specific autoantibodies [279,280,281,282]. Then, the autoantibodies trigger cross-reactive immunogenicity against basement membrane proteins BP180 and BP230 to induce bullous pemphigoid [279,281,282]. Furthermore, in T cell-dependent humoral immunity, PD-1 acts as an activator for B cells and enables their interaction with either T helper cells or Treg cells within the germinal centers [283,284]. T helper cells play a role in the selection and survival of B cells and enable their differentiation into memory B cells or high-affinity antibody-producing plasma cells. While Treg cells maintain immune balance by suppressing both T helper cells and B cells [285,286]. Inhibition of PD-1 may reduce the suppressive ability of Treg cells and lead to potentially mutated B cells selected by T helper cells [281,287]. As a result, abnormal production of low-affinity plasma cells occurs, which contributes to the development of numerous antibody-mediated autoimmune diseases, including bullous pemphigoid [288,289]. In addition to cross-reactivity, the production of autoantibodies against different epitopes, the so-called epitope spreading phenomenon, has been observed in anti-PD-1/PD-L1-associated bullous pemphigoid [281,290].

### 6.6. Vitiligo 

Vitiligo is a chronic, acquired dyschromia that promotes autoimmune aggression against melanocytes, ultimately resulting in the formation of hypochromic or achromic patches and dots on the skin and mucous membranes [13,275]. The pathogenesis of vitiligo or vitiligo-like depigmentation is multifactorial and includes individual genetic susceptibility, melanocyte auto-aggression, and failure of immune tolerance mechanisms [55,58,59]. The development of vitiligo occurs most frequently in melanoma patients after treatment with ICIs and is less common in patients treated for other cancers [66,67]. Vitiligo development occurs in melanoma patients receiving anti-CTLA-4 and/or anti-PD-1 inhibitors as monotherapies or combination therapies. Several studies have indicated that patients receiving combinational therapy with CTLA-4, and PD-1 inhibitors are more likely to develop vitiligo compared to those receiving monotherapy [66,67]. The development of vitiligo during or after ICIs treatment is clinically significant, as the appearance of vitiligo not only indicates immune system activation but also has positive prognostic implications in melanoma patients [291,292]. The paradoxical nature of the association between vitiligo and improved outcome suggests the possibility that depigmentation is attributed to a strong antitumor immune response [293,294]. Conversely, another aspect to be considered is the timing of vitiligo onset in relation to ICI therapy. While some studies report the development of vitiligo-like depigmentation just a few weeks after the treatment has begun, other studies report a delayed onset of the disease by several months after starting therapy [295,296]. These large differences make it necessary to ensure continuous monitoring of patients undergoing ICIs. To date there is no definitive treatment for ICI-associated vitiligo. Ongoing photoprotection with clothing and sunscreen is the best option to avoid sunburns in these patients [238,297]. While there is no specific medication to stop ICI-induced vitiligo, there are some drugs that can slow pigmentation loss [283,298]. The new vitiligo medication ruxolitinib (Opzelura) is the most promising class of drug that inhibits both JAK 1 and 2, leading to a significant improvement in facial vitiligo showing a significant improvement [188,266,299]. Although there is increasing evidence suggesting a possible link between immune-related adverse effects and clinical benefit in some cancers [300,301,302,303]. Consequently, the more development of immune-related adverse effect, the better response of cancer patients to ICI therapy is expected. The relationship between cutaneous adverse reactions and treatment efficacy has been the subject of extensive research, particularly in the context of ICIs in various malignancies. Several studies have investigated the relationship between the timing of cutaneous adverse effects onset and treatment efficacy in patients with different tumor types treated with PD-1/PD-L1 inhibitors [300,301,302,303]. The relationship between immune-related adverse events and treatment efficacy in patients with NSCLC has been investigated in several studies [304,305,306]. To evaluate the relationship between immune-related adverse effects occurrence and treatment outcomes with atezolizumab [307], pooled data from three Phase 3 clinical trials, namely, IMpower130 [308], IMpower132 [309], and IMpower150 [310] have been analyzed. The most common immune-related adverse effects were skin rash (28%), liver dysfunction (15%), and thyroid dysfunction (12%) [303]. However, patients with immune-related adverse effects exhibited longer overall survival (OS) when compared with the treated patients, who exhibited no immune-adverse effects [306,311,312]. Additionally, in melanoma patients the development of immune-related adverse effects was correlated with treatment efficacy [312,313,314]. The incidence of immune-related adverse effects among nivolumab-treated patients was 68.2%, and patients with immune-related adverse effects exhibited a marked difference in overall survival, when compared with treated patients, who do not exhibit immune-related adverse effects [315,316]. However, the most significant association was noted between immune-related adverse effects such as rash and vitiligo and overall survival in patients with metastatic disease [303,317].

### 6.7. Pruritus 

Pruritus is a common skin symptom occurring in many diseases. It is an unpleasant feeling on the skin and often affects the quality of life of patients [318,319]. The most common causes include contact with an allergen, dry skin, and a reaction to medication. Pruritic rash is associated with ICI-based therapy [320,321], with the highest occurrence manifesting in response to anti-CTLA-4 inhibitors alone or in combination with an anti-PD-1 inhibitor [213,322,323], While pruritus is usually associated with the development of other skin diseases, its development in cancer patients can occur independently from cutaneous alterations [324,325,326], either as a direct or indirect side effect of ICIs. Although pruritus is typically mild, a small portion of patients develop severe pruritus. Intense or widespread pruritus symptoms cannot be well controlled with available treatment [327,328,329,330]. In severe cases, symptoms can severely impair quality of life, even following discontinuation of ICIs [325,326,327,328,329,330]. For grade 1/2 pruritus, moderate to high potency topical corticosteroids, oral antihistamines, and topical emollients are recommended; immunotherapy can usually be continued [223,224,225,226,227,228]. The most common treatments of ICI-associated cutaneous adverse effects included topical corticosteroids up to 80% of patients; systemic corticosteroids and immunomodulators up to 60% of patients, while discontinuation or cessation of immunotherapy up to 3.9% of patients [227,228,229,230,231,232,233,234,235,236,237,238,239,240,241,242,243,244,245,246,247,248,250,251,252,253,254,255,256,257,258,259,260,261,262,263,264,265,266,267,268,269,270,271,272,273,274,275,276,277,278,279,280,281,282,283,284,285,286,287,288,289,290,291,292,293,294,295,296,297,298,299,300,301,302,303,304,305,306,307,308,309,310,311,312,313,314,315,316,317,318,319,320,321,322,323,324,325,326,327,328,329,330]. Although mild cases of cutaneous adverse reactions have been shown to resolve within 3 months, standard treatment for mild skin rashes has consisted of topical corticosteroids or urea-containing creams and antihistamines with continued immunotherapy [330]. Severe rashes were treated by discontinuation of ISIs and administration of systemic corticosteroids [330].

## 7. Conclusions

ICIs represent a breakthrough in cancer therapy based on their promising efficacy in the treatment of many solid and hematologic malignancies. Although ICI-based therapies have a favorable risk–benefit ratio and a specific safety profile, their unique mechanism of action is mostly associated with variable adverse effects. These adverse effects can impact both quality of life and treatment efficacy through dose limitations or discontinuation. ICI-associated cutaneous adverse effects are the most frequent adverse events observed to date. The dermatologic safety profiles of CTLA-4 and PD-1/PD-L1 inhibitors are quite similar, and their combined use is associated with significant increases in cutaneous adverse effects. The most common ICI-induced cutaneous adverse effects include nonspecific maculopapular rashes, vitiligo, lichenoid dermatitis, bullous dermatosis, psoriasiform dermatitis, and pruritus. Fortuitously, cutaneous side effects caused by PD-1/PD-L1 or CTLA-4 blockade are self-limiting and easily manageable. However, all severe, persistent, or atypical lesions require a comprehensive dermatological examination. Early detection and appropriate treatment are crucial in reducing dose-limiting toxicities.

## Figures and Tables

**Figure 1 ijms-26-00088-f001:**
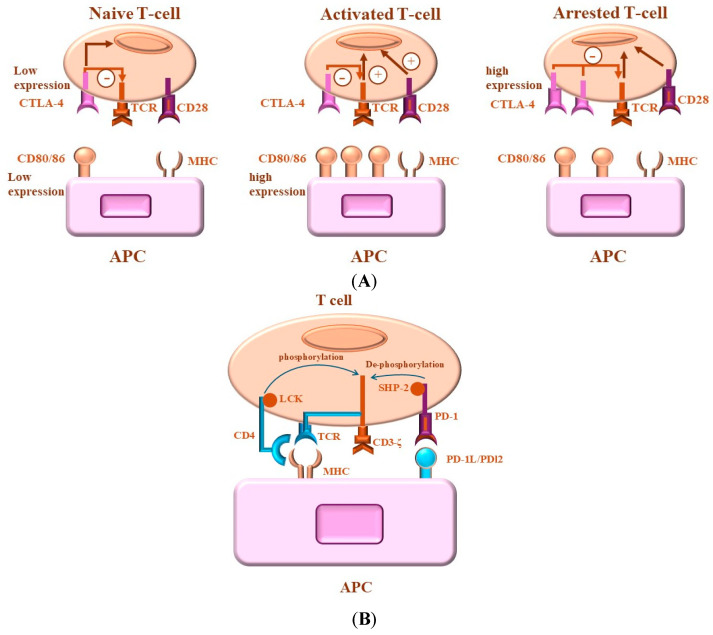
Proposed models of CTLA-4- and PD-1-mediated regulation of T cells. (**A**) CTLA-4-mediated T-cell activation. Naive T-cell activation is mediated by both TCR and CD28 signaling. CTLA-4 expression results in the inhibition of activated T cells by directly competing with CD28 for ligand binding and/or by generating inhibitory signals. Whereas suboptimal co-stimulation of T cells and CTLA-4 expression leads to direct competition between CD28 and CTLA-4 for ligand binding, optimal co-stimulation is essential for limiting full T-cell activation. The successful competition of CTLA-4 against CD28 mediates inhibitory signals, which terminate the T-cell response. (**B**) Programmed Death (PD)-1-mediated regulation of T cells. The recognition of the MHC–antigen complex by the T-cell receptor (TCR) and CD4 leads to LCK kinase-mediated phosphorylation of the CD3-TCRζ complex. PD-1 ligation by its ligands brings PD-1 close to TCR, allowing the SHP-2 phosphatase to dephosphorylate the CD3-TCRζ complex and attenuate the signal.

**Figure 2 ijms-26-00088-f002:**
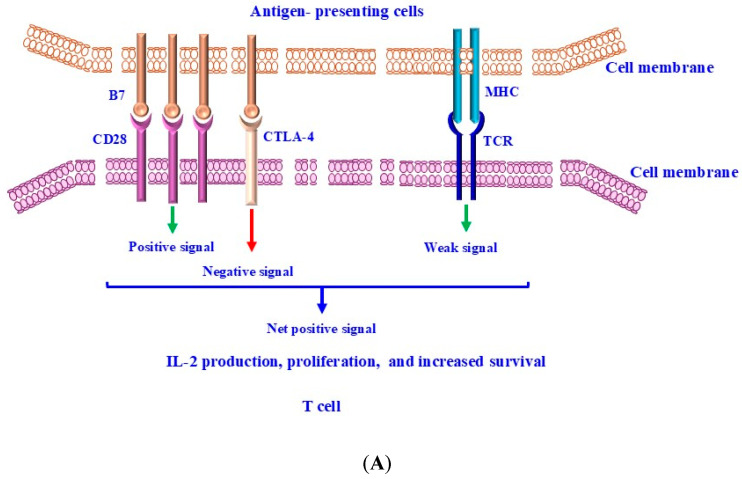
CTLA-4-mediated inhibition of T cells. T cells are activated when TCRs bind antigens, which are presented by MHC on APCs, and by CTLA-4:B7- and CD28:B7-mediated stimulation. (**A**) The weak T-cell activation-dependent TCR stimulus is mediated by CD28:B7 binding, leading to the transduction of positive T-cell-intrinsic signaling, and by CTLA-4:B7 binding, leading to the transduction of negative T-cell-intrinsic signaling. MHC: TCR binding leads to the transduction of weak T-cell-intrinsic signaling. All transduced signals result in the generation of a net positive signal, which leads to IL-2 production and enhanced proliferation and survival of T cells. (**B**) T-cell activation-dependent strong TCR stimulus. CTLA-4 expression is upregulated by increased transport from intracellular stores to the cell surface and reduced internalization. CTLA-4 competes with CD28 for binding of B7 molecules. Increased CTLA-4:B7 binding may result in a net negative signal that limits IL-2 production and proliferation while restricting T-cell survival. CTLA-4 indicates cytotoxic T-lymphocyte-associated antigen 4; IL-2: interleukin-2; MHC: major histocompatibility complex; TCR: T-cell receptor. Red arrow: 
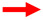
 means negative signal; while green arrow: 
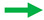
 means weak or positive siganal.

**Figure 3 ijms-26-00088-f003:**
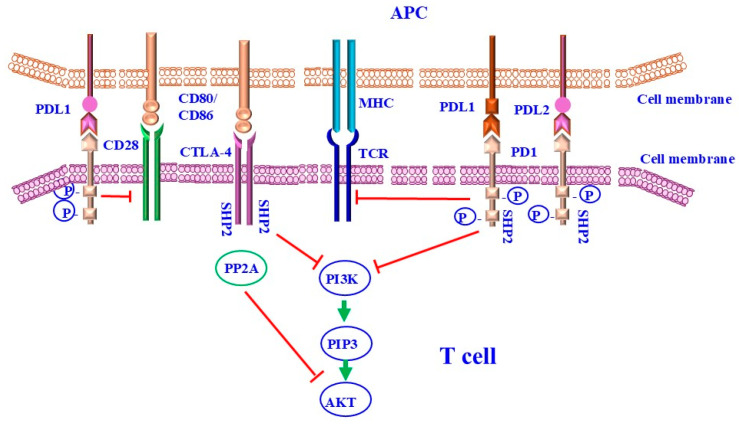
Mechanisms of CTLA-4 and PD-1 signaling-mediated T-cell inhibition. CTLA-4 and PD-1 negatively regulate T-cell activation. CD28 mediates TCR and MHC attachment. CTLA-4 acts as a competitive homolog to CD28 and binds to CD80/CD86, ligands of CD28, thereby preventing T-cell activation. Both the CTLA-4 and PD-1 pathways are activated via TCR-dependent activation. CD28 receptor is a relevant target of PD-1-mediated inhibition. PD-1 binding to PD-L1 also negatively regulates T-cell activation through the recruitment of SHP-CTLA-4, and PD-1 recruited SHP-2 inhibits PI3K downstream signaling. CTLA-4 reacts with PP2A to dephosphorylate AKT, leading to inhibition of T-cell activation. PD-1:PD-L1 binding inhibits TCR-mediated positive signaling, leading to reduced proliferation of T cells. SHP-2: Src homology region-2 containing protein tyrosine phosphatase; PP2A: serine/threonine phosphatase PP2A.

**Figure 4 ijms-26-00088-f004:**
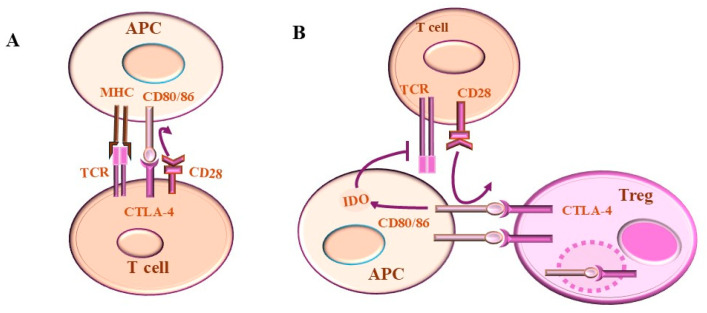
The role of CTLA-4 in regulating T-cell activation and Treg-cell function. (**A**) CTLA-4 interacts with CD80/86 on APCs to compete with CD28 for ligands. CTLA-4 is characterized by a higher binding affinity for CD80/86 than those of CD28 and thereby CTLA-4 can block the interaction of CD80/CD86 with CD28. (**B**) CTLA-4 on Treg cells binds to CD80/86 on APCs, blocks costimulatory signaling in conventional T cells, and depletes CD80/86 by trans-endocytosis. Therefore, CD28 of conventional T cells cannot interact with CD80/86, which may lead to a reduction in T-cell activation. The ability of CTLA-4 to induce indolamine 2, 3-dioxygenase (IDO) from antigen presenting cells (APCs) is a mechanism whereby the CTLA-4 triggers T-cell inhibition.

**Figure 5 ijms-26-00088-f005:**
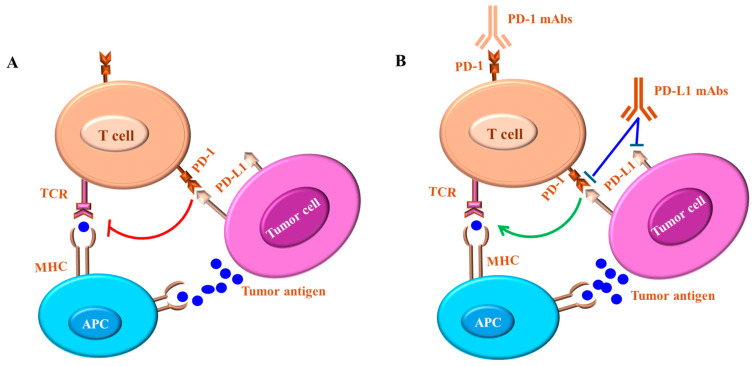
Mechanisms of PD-1/PD-L1 pathway-dependent tumor immune escape. (**A**) Binding of PD-1 to PD-L1 on the surface of immune effector cells (T cells) suppresses T cell receptors (TCR) in recognizing the major histocompatibility (MHC) molecules on the surface of antigen presenting cells (APCs)/tumor cells. (**B**) The inhibition of PD1 binding to PD-L1 by either anti-PD-1 or PD-L1 antibodies enhances the activation of T cells through binding of TCR to MHC on the surface of APCs to trigger tumor cell death.

**Figure 6 ijms-26-00088-f006:**
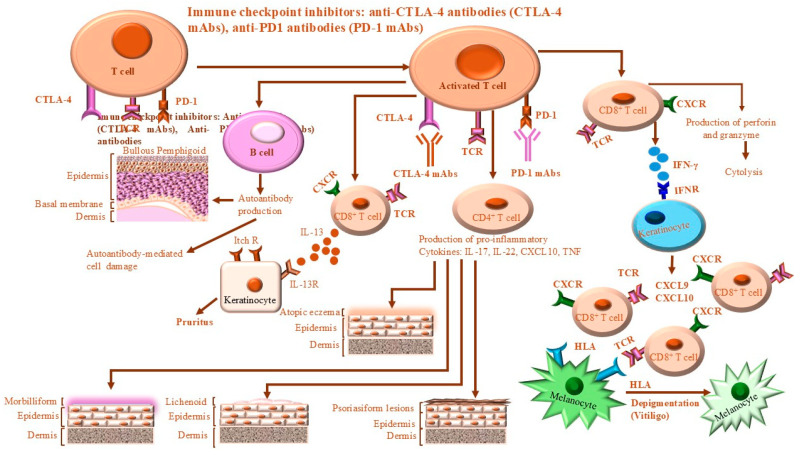
Immune checkpoint inhibitors: anti-CTLA-4 antibodies (CTLA-4 mAbs), anti-PD1 antibodies (PD-1 mAbs) block CTLA-4 and PD-1 receptors on the surface of T cells. ICI-induced accumulation of activated CD8^+^ T-cell population is characterized by T-cell receptors (TCR) and chemokine receptors (CXCR) expression and the production of perforin, granzymes, and cytokines such as interferon gamma (IFNγ), interleukin (IL)-4, IL-9, IL-13, IL-17, and IL-22. ICI-induced accumulation of activated CD4^+^ T cells is characterized by the production of pro-inflammatory cytokines including IL-17, IL-22, CXCL10, and TNF. ICI-induced activation of B cells is associated with the production of autoantibodies associated with bolus pemphigoid and autoantibody-mediated cell damage. Mechanisms of ICI-induced vitiligo: The activation of CD8^+^ T-cell populations by ICIs results in a release of IFNγ that can enhance keratinocyte activation to produce the chemokines CXC9 and CXC10, which ultimately recruit CD8^+^ T cells to the epidermis to enhance melanocyte loss and allow vitiligo development. Mechanisms of ICI-induced pruritus: The production of IL-13 by CD8^+^ T cells enhances keratinocyte activation which increases the expression of itch-associated receptors and pruritus development. Mechanisms of ICI-induced atopic eczema, psoriasiform lesions, lichenoid, and morbilliform are mediated by the pro-inflammatory cytokines IL-17, IL-22, CXCL10, and TNF released by ICI-induced CD4+ T-cell activation. Mechanisms of ICI-induced bullous pemphigoid: The blockade of PD-1/PDL-1 by ICIs in T cells enhances B-cell activation, leading to the production of disease-specific autoantibodies. These autoantibodies include IgG and IgE, which induce subepidermal blistering via direct and indirect mechanisms. The cross-reactive immunogenicity of IgG4 against the basement membrane proteins BP180 and BP230 leads to the direct formation of subepidermal blistering. IgE induces the formation of subepidermal blistering via the enhancement of mast cell degranulation and the recruitment of eosinophiles and neutrophiles, which release proteolytic enzymes. IgG1 also induces subepidermal blistering via complement activation-mediated mast cell degranulation to recruit eosinophiles and neutrophiles.

**Table 1 ijms-26-00088-t001:** Systemic agents used for the treatment of ICI-induced cutaneous adverse effects.

Systemic Treatment	Indications	Dose	Comments
Anti-IL6 [184]	Maculopapular rash, lichenoid rash	Tocilizumab 162 mg/2 weeks.	For corticosteroid-resistant maculopapular or lichenoid rash
Dupilumab [185,186]	Bullous pemphigoid, eczematous dermatitis	300 mg/every other week	Dupilumab treatment carries lower risks of systemic immunosuppression
Omalizumab [187]	Bullous pemphigoid, eczematous dermatitis, urticaria, pruritus	300 mg/month	Immunoglobulin E (IgE) blocker
Rituximab [188]	Bullous pemphigoid	375 mg/m2 once weekly for 4 weeks	Anti-CD20 antibodies
IVIG [189]	Dermatomyositis, bullous pemphigoid, SJS/TEN	BP: IVIG 1–2 g/kg every 4 weeks	Safe to administer in the context of malignancy
Apremilast [190]	Psoriasiform	30 mg twice daily	Mostly interacts with the innate immune system and is considered a relatively safe option for cancer patients.

## Data Availability

No new data were created or analyzed in this study.

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
