# Peer review of "Immune Checkpoint Inhibitor-Associated Cutaneous Adverse Events: Mechanisms of Occurrence"

_ijms, 2024, doi:10.3390/ijms26010088_

Round 1

Reviewer 1 Report (New Reviewer)

Comments and Suggestions for Authors

Dear authors,

The manuscript addresses an important and clinical relevant topic. However, a series of revisions are required.

·       “The most observed ICIs-induced adverse effects are mediated by the activation of autoreactive T-cells leading to the occurrence of various adverse effects like autoimmune diseases. As well as gastrointestinal, endocrine, and dermatologic toxicity [5-12]. ICIs treatment  is associated with dermatological [13, 14], gastrointestinal [15, 16], , pulmonary [7, 17] , renal [18-20] , ophthalmologic [21, 22], rheumatic [23, 24], cardiovascular [25], and hematologic [26], 51 adverse events with varying frequencies and severities.” (lines 47-52) Please rephrase this paragraph in order to avoid repetitiveness.

·       Please explain abbreviations and acronyms at their first use in the text.

·       “Also, the treatment of cancer patients with anti CTLA-4 (ipilimumab) can result in the development of other cutanous advers effects such as neutrophilic dermatoses, pustular eruptions, bullous lupus erythematosus, and pyoderma gangrenosum [ 31,32].” (lines 65-67) Please refer to cutaneous adverse effects triggered by anti-PD1 and anti-PDL1 agents apart from the  maculopapular rash, as well.

·       “…innate cells express a set of germline-encoded receptors which play roles in adaptive immune responses [46-48]. (lines 99-100) Please explain or rephrase

·       “CTLA-100 4 is mainly expressed on activated T- cells and Treg [41]…” (line 100) Please remove as this has already been stated in the same paragraph.

·       “Both T and B lymphocytes play a crucial role in the regulation of the acquired immune response and hematopoietic homeostasis.” (lines 125 - 126) T and B cells are the drivers of the adaptive immune response, exerting various effector functions. Tregs and Bregs are involved in the regulation of the adaptive response. Please rephrase for better clarity.

·       “As is known, the activation of resting T cells is a complex process that is tightly regulated by intracellular signaling pathways, the initiation of which is mediated by the T cell antigen receptor (TCR) through costimulatory and negative receptor signaling [ 66, 67].” (lines 126-129) I suggest rephrasing to “the blockade of inhibitory receptor signaling”.

·       “These autoantibodies form  immune complexes involved in the development of several types of autoimmune diseases [156, 326 157] and bullous pemphigoid [158].” (lines 325-327) Please consider rephrasing. Bullous pemphigoid is also an autoimmune disease. Moreover, it is not the only autoimmune blistering disease that may develop in these patients.

·       “Upon ICIs treatment, the inflammatory reaction may occur in response to cytokine and other effector immune cell release [164, 165].” (lines 334-335) Please provide examples of such cutaneous reactions.

·       “Eczematous dermatitis, also known as atopic dermatitis or eczema, is a chronic inflammatory  skin condition that causes dry, itchy, red skin.” (line 384) Atopic dermatitis is a form of chronic eczema.  The influence of ICIs on atopic dermatitis should be discussed or the mention of atopic dermatitis removed.

·       “The development of eczematous dermatitis during immunotherapy treatment is one of the most common cutaneous adverse effects occurring through PD-1 inhibitor-dependent mechanisms [63, 64]. Treatment of cancer patients with CTLA-4-based therapy is therefore associated with eczematous dermatitis occurrence [63, 391 65].” (lines 388-390) Please provide epidemiologic data. The frequency of eczema occurring during ICI treatment should be provided.

·       “In conclusion, while the interruption or discontinuation of ICI therapy in patients with  morbilliform maculopapular rash is rare, in severe cases ICI therapy can be discontinued following treatment with systemic steroids.” (lines 397-399) Confusion between eczema and morbiliform or maculopapular rashes exists. Please correct.

·       “In severe eczematous dermatitis, systemic corticosteroids and NB-UVB phototherapy are often the most beneficial treatment modalities [188, 189].” (lines 403-404) The doses of systemic corticosteroids and duration of corticotherapy recommended in this cathegory of patients should be discussed.

·       The management of patients developing eczema during ICI treatment should be stratified according to the severity of this side effect.

·       Both “eczematous dermatitis” and “morbiliform maculopapular rash” are pleonasms. Please rephrase

·       “Other symptoms include mottling or plaques, fever, runny nose, and bleachable rash areas  [190-192]. As the mucous membranes are typically spared, it is possible to distinguish between mucous membranes less and more erupted [190, 191].” (lines 409-411) The mention of a “runny nose” should be removed as the next line states mucosae are spared. Please explain the meaning of “mucous membranes less and more erupted”.

·       “The appearance of a morbilliform  (maculopapular) rash on the trunk and extremities was noted in tumor patients following treatment with anti-CTLA-4 therapy [189-191].” (lines 411-412) The maculopapular rash is also encountered in patients receiving anti-PD1 or anti-PDL1 treatment.

·       “Of note, most rashes caused by immunotherapy are associated with a morbilliform reaction [189-191].” (line 413) Please provide details regarding the frequency of this type of reaction.

·       “Eruptions are typically mild and resolve spontaneously two to three months after treatment with anti-CTLA-4 and/or anti-PD-1 inhibitors. [189, 415 192]. “ Please refer to the course of the skin rash in patients continuing ICIs.

·       “Newer ICIs targeting PD-L1 are associated with less frequent cutaneous adverse events than  those targeting either CTLA-4 or PD-1 receptors [63, 65].” (line 416) Please provide epidemiologic data.

·       “The appearance of morbilliform maculopapular rash is attributed to heterogeneous  mechanisms which regulate measles-like rashes following immunotherapy.” (line 418) Please present this mechanisms in detail.

·       “Identified rashes in  immunotherapy treated cancer patients appear either as itchy, bleaching, erythematous patches or  papules on most of the body surface, especially on the head, palms and the soles of the feet [191, 421 192].” (lines 419-420) This phrase should be removed as it is repetitive

·       “Thus, patients should be carefully examined for blistering, mucous membrane inflammation,  epidermal detachment, high fever, and/or swollen lymph nodes several days to one week after the onset of such rashes.” These are sign of Stevens Johnson syndrome or toxic epidermal necrolysis, which are not mentioned. They do not progress from maculopapular rashes, but should be considered in the differential diagnosis.

·       The management of patients developing maculopapular rash during ICI treatment should be stratified according to the severity of this side effect.

·       “Lichen planus often appears as purple, itchy, flat bumps that develop over several weeks.” (line 429) Please provide a more accurate description of lichen planus skin and mucosal lesions, using academic terms.

·       “The onset of rashes can occur either immediately or after the start of treatment, especially in patients undergoing  treatment with PD-1 inhibitors [192, 193].” (lines 440-441) Please provide more precise data on the possible duration between the start of ICI therapy and the appearance of lichen planus, on the frequency of this side effect, on the influence of preexistent lichen planus and the mechanisms underlying this association.

·       “Psoriasis is a skin disorder that causes skin irritation and the formation of scaly, red plaque- like psoriasis [186, 212].” (line 453) This description is incomprehensive.

·       “The development of psoriasis in cancer patients is one of the common cutaneous side effects observed for patients following ICI treatment [214, 215].” (lines 455-457) This is inaccurate. Please provide epidemiologic data and the mechanisms underlying this association.

·       “Cancer patients with pre-existing psoriasis have shown improvement during ICI therapy, suggesting a complex interaction between the immune system and these medications [218, 219].” The contrary has also been observed. 

·       The frequency of bullous disorders induced by ICI treatment, the influence of this treatment on preexistent bullous diseases and the mechanisms underlying ICI-induced bullous disorders should be thoroughly presented.

·       The possible correlation between cutaneous adverse effects and ICIs efficacy should be more thoroughly discussed.

·       The effect of systemic corticosteroids and immunosuppressive agents used to control the cutaneous side effects, as well as the optimal management of these patients should be addressed

·       English language editing is required.

Best regards!

Comments on the Quality of English Language

English editing is required.

Author Response

Comments and Suggestions for Authors
Dear Editor,
Thank you very much for the helpful comments of the reviewer. As requested, we answered the suggested comments of the reviewer point-for-point.
Now, we hope the manuscript is improved and meets the quality of the publication.
On behalf all my co-authors 

The manuscript addresses an important and clinically relevant topic. However, a series of revisions are required.
Comment: “The most observed ICIs-induced adverse effects are mediated by the activation of autoreactive T-cells leading to the occurrence of various adverse effects like autoimmune diseases. As well as gastrointestinal, endocrine, and dermatologic toxicity [5-12]. ICIs treatment  is associated with dermatological [13, 14], gastrointestinal [15, 16], , pulmonary [7, 17] , renal [18-20] , ophthalmologic [21, 22], rheumatic [23, 24], cardiovascular [25], and hematologic [26], 51 adverse events with varying frequencies and severities.” (lines 47-52) Please rephrase this paragraph in order to avoid repetitiveness.
Authors response: Thank you very much for your comment. Accordingly, we rephrased he paragraph and avoided the reptation
See lines:47-56; the following paragraph [The most observed ICIs-induced adverse effects are mediated by the activation of autoreactive T-cells leading to the occurrence of various adverse effects like autoimmune diseases. As well as gastrointestinal, endocrine, and dermatologic toxicity [5-12]. ICIs treatment is associated with dermatological [13, 14], gastrointestinal [15, 16], , pulmonary [7, 17] , renal [18-20] , ophthalmologic [21, 22], rheumatic [23, 24], cardiovascular [25], and hematologic [26], adverse events with varying frequencies and severities.    ] has been modified and replaced by the following paragraph [.The most observed ICIs-induced adverse effects are mediated by the activation of autoreactive T-cells leading to the occurrence of various adverse effects like autoimmune diseases. As well as gastrointestinal, endocrine, and dermatologic toxicity [5-12]. ICIs treatment is associated with dermatological [13, 14], gastrointestinal [15, 16], , pulmonary [7, 17] , renal [18-20] , ophthalmologic [21, 22], rheumatic [23, 24], cardiovascular [25], and hematologic [26], adverse events with varying frequencies and severities. has been modified and replaced by the following paragraph [The most commonly observed adverse reactions induced by ICIs are mediated by the activation of autoreactive T cells, leading to the occurrence of various adverse reactions such as autoimmune diseases [5-12], as well as gastrointestinal [15, 16], dermatological [13, 14], pulmonary [7, 17], renal [18-20], ophthalmological [21, 22], rheumatic [23, 24], cardiovascular [25] and hematological [26] adverse reactions with varying frequency and severity.  ] 
Comment: Please explain abbreviations and acronyms at their first use in the text.
Authors response: Thank you very much for your comment. As required explained abbreviations at their first use.
Comment: Also, the treatment of cancer patients with anti CTLA-4 (ipilimumab) can result in the development of other cutanous advers effects such as neutrophilic dermatoses, pustular eruptions, bullous lupus erythematosus, and pyoderma gangrenosum [ 31,32].” (lines 65-67) Please refer to cutaneous adverse effects triggered by anti-PD1 and anti-PDL1 agents apart from the  maculopapular rash, as well.
Authors response: Thank you for your valuable commen. We rereferred also to the cuanous adverse effect caused by anti-PD-1 and anti-PD-L1.
The following modifications has been made in the manuscript
Text
See Lines.67-81, the following paragraph [The therapeutic use of humanized monoclonal ICIs of PD-1 (pembrolizumab and nivolumab) and PD-L1 (atezolizumab, avelumab, durvalumab) as effective cancer therapies is widely established [31, 32].
The signaling mechanism of anti-PD-1/PD-L1 involves triggering the activation of cytotoxic CD4+/CD8+ T cells and subsequent killing of cancer cells, causing specific immunological side effects specific to the inhibitors of both PD-1 and PD-L1 [33, 34] These medications can cause numerous skin reactions and are considered the most common immune-related adverse reactions. Most cutaneous adverse effects range from nonspecific rashes to recognizable skin lesions that sometimes resolve spontaneously and have an acceptable skin toxicity profile. However, some may cause life-threatening complications [35, 36] 
Blockade of the PD-1/PD-L1 pathway has been observed to increase B-cell activation, proliferation, and subsequently production of disease-specific autoantibodies including anti- bullous pemphigoid 180 (BP180) antibody leading to development of bullous pemphigoid [37] . The development of vitiligo has been reported in patients with melanoma after treatment with PD-1 and PD-L1 inhibitors. The occurrence of lichenoid reactions, pruritus and measles-like eruptions was observed to a low degree, so that immediate discontinuation of therapy was not urgent [38, 39 ] ] has been added to the main text of the manuscript 
Comment:  “…innate cells express a set of germline-encoded receptors which play roles in adaptive immune responses [46-48]. (lines 99-100) Please explain or rephrase
Authors response: Thank you very for your comment. We modified the text to be more informative 
See lines:100-112; the following sentences [ Immune dysregulation is maladaptive change in molecular control of immune system through various processes mediated by ICIs, component in the pathogenesis of autoimmune diseases and cancers-dependent mechanisms. CTLA-4 immune is a checkpoint molecule that is expressed on activated T- cells and regulatory T- cells (Treg), which inhibit T- cell activation and regulate immune homeostasis. The adaptive immune system is not the only mediator involved in anti-tumor immunity regulation [41-43] [.. Many innate leukocytes can distinguish normal cells from tumor cells and exert tumor-suppressive functions [44, 45] [. . While conventional T- cells recognize cancer cells using a rearranged antigen receptor with countless specificities for tumor antigens, innate cells including phagocytic cells, epithelial and endothelial cells, NK cells, innate lymphoid cells, and platelets are characterized by the expression of a fixed set of germline-encoded receptors that are known as pattern recognition receptors (PRRs) [46, 47]. These types of receptors play an important role in the regulation of adaptive immune responses [55-57] Since the enhancement of the adaptive response is based on recognition mechanisms of innate immune cells [ 50, 51. 
The expression of CTLA-4 in activated T- cells and Treg cells has been reported to paly a crucial in the regulation of the therapeutic efficacy of ICIs in the treatment of autoimmune diseases and cancer [41,52].] [innate cells express a set of germline-encoded receptors which play roles in adaptive immune responses [46-48]. have been modified and replaced by the following text [  ]
References
Comment: “CTLA-4 is mainly expressed on activated T- cells and Treg [41]…” (line 100) Please remove as this has already been stated in the same paragraph.
Authors response: Thank you very much for your comment. As required, we removed the mentioned sentence 

Comment: Both T and B lymphocytes play a crucial role in the regulation of the acquired immune response and hematopoietic homeostasis.” (lines 125 - 126) T and B cells are the drivers of the adaptive immune response, exerting various effector functions. Tregs and Bregs are involved in the regulation of the adaptive response. Please rephrase for better clarity.
Authors response: Thank you very much for your comment. As required, we rephrased the sentence as following
Text
See lines:131-134 the following sentence [ Both T and B lymphocytes play a crucial role in the regulation of the acquired immune response and hematopoietic homeostasis. ] has been rephrased and replaced by the following sentence [While antigens expressed on tumor cells or on cells suffering from other pathological diseases are recognized by the immune system, the presentation of these antigens alone is not sufficient to trigger an effective immune response [49, 66]. T and B cells are key components of the adaptive immune system. Through their immune properties and their interactions with other immune cells and cytokines in their environment, they form a complex network to achieve immune tolerance and maintain the body's homeostasis.]

Comment: As is known, the activation of resting T cells is a complex process that is tightly regulated by intracellular signaling pathways, the initiation of which is mediated by the T cell antigen receptor (TCR) through costimulatory and negative receptor signaling [ 66, 67].” (lines 126-129) I suggest rephrasing to “the blockade of inhibitory receptor signaling”.
Authors response: Thank you very much for your valuable comment.  As required, modified the sentences
Text
See lines:134-137; the following sentences [As is known, the activation of resting T cells is a complex process that is tightly regulated by intracellular signaling pathways, the initiation of which is mediated by the T cell antigen receptor (TCR) through costimulatory and negative receptor signaling [ 66, 67] ] have been modified and replaced by the following Text [As is known, the  activation of naïve T- cells is mediated by two signals. The first one is derived from TCR that recognizes a small part of the antigen to ensure the specificity of the response. So that only T- cells that recognize this antigen will be activated. While the second signal that is known as co-stimulatory signal is provided by the co-stimulatory molecule, CD28 [67, 68]. ]   
Comment: “These autoantibodies form  immune complexes involved in the development of several types of autoimmune diseases [156, 326 157] and bullous pemphigoid [158].” (lines 325-327) Please consider rephrasing. Bullous pemphigoid is also an autoimmune disease. Moreover, it is not the only autoimmune blistering disease that may develop in these patients.
Authors response: hank you very much for your comment. As suggested, we rephrased the mentioned sentence
See lines: 302-303 ; the following sentences [ These autoantibodies form immune complexes involved in the development of several types of autoimmune diseases [156, 157]. and bullous pemphigoid [158].  ] have been rephrased to the following sentences [These autoantibodies form immune complexes involved in the development of several types of autoimmune diseases [156, 168], including  bullous pemphigoid [157]..] 

Comment:Upon ICIs treatment, the inflammatory reaction may occur in response to cytokine and other effector immune cell release [164, 165].” (lines 334-335) Please provide examples of such cutaneous reactions.
Authors response: Thank you very much for your comment. As required, we added examples for the ICIs-associated inflammatory reaction.
See lines: 310-313; the following paragraph [ICIs-associated inflammatory skin reactions that is known as a group of several skin rashes including maculopapular, lichenoid, psoriasiform and eczematous rashes as well as cutaneous adverse like erythema multiforme, palmoplantar erythrodysesthesia and neutrophilic dermatoses, such as Sweet syndrome [29, 163-165].has been added.

Comment: “Eczematous dermatitis, also known as atopic dermatitis or eczema, is a chronic inflammatory skin condition that causes dry, itchy, red skin.” (line 384) Atopic dermatitis is a form of chronic eczema.  The influence of ICIs on atopic dermatitis should be discussed or the mention of atopic dermatitis removed.
Authors response: Thank you for your comment. Accordingly, we removed “atopic dermatitis”
Comment: “The development of eczematous dermatitis during immunotherapy treatment is one of the most common cutaneous adverse effects occurring through PD-1 inhibitor-dependent mechanisms [63, 64]. Treatment of cancer patients with CTLA-4-based therapy is therefore associated with eczematous dermatitis occurrence [63, 391 65].” (lines 388-390) Please provide epidemiologic data. The frequency of eczema occurring during ICI treatment should be provided.
Authors response: thank you very much for your comment. As required, we provided epidemiological data about the of the frequent occurrence of eczema during ICI treatment.
See lines:355-358; the following text [The incidence of developing eczema has been reported to increase at an approximately constant rate after the treatment of ICIs [64, 194] .. A cumulative incidence rate of 25% was shown to develop first eczema within 10.3 months, a cumulative incidence rate of 33% of patient within 13.8 months and a cumulative incidence rate of 40% within 20 months after the treatment has been initiated [64.].] has been added to the text 
Comment: “In conclusion, while the interruption or discontinuation of ICI therapy in patients with  morbilliform maculopapular rash is rare, in severe cases ICI therapy can be discontinued following treatment with systemic steroids.” (lines 397-399) Confusion between eczema and morbiliform or maculopapular rashes exists. Please correct.
Authors response: thank you very much for your comment. Accordingly, we corrected the misake
See line: 376; the flowing sentence [ A morbilliform maculopapular rash, also known as an exanthematous or maculopapular drug rash, is a measles-like rash formed .. ]  has been rephrased as following [A morbilliform, also known as maculopapular drug reruption ] .
Comment : “In severe eczematous dermatitis, systemic corticosteroids and NB-UVB phototherapy are often the most beneficial treatment modalities [188, 189].” (lines 403-404) The doses of systemic corticosteroids and duration of corticotherapy recommended in this cathegory of patients should be discussed. The management of patients developing eczema during ICI treatment should be stratified according to the severity of this side effect.
Authors response: Thank you very much for your comment. As required, we added more information about the treatment options for sever eczematous dermatitis without to refer to the recommended  doses and duration. Since we believe this review is target for  reader with no clinical experience or background in the field dermatology.
Text 
See lines:366-372; the following sentences [In severe eczematous dermatitis, systemic corticosteroids and NB-UVB phototherapy are often the most beneficial treatment modalities [188, 189] ] were replaced by the following paragraph [Although environmental interventions together with topical devices including emollients, corticosteroids, and calcineurin inhibitors represent the mainstay of treatment, systemic treatments are reserved for severe cases [187, 188] [. Phototherapy represents a valid second-line intervention in those cases where non-pharmacological and topical measures have failed. Different forms of light therapy are available and have showed varying degrees of beneficial effect [189, 190] . The most common phototherapeutic options of severe eczematous dermatitis are the natural sunlight, narrowband (NB)-UVB, broadband (BB)-UVB, UVA, UVA1 that can administrated either as monotherapy or in combination with systemic corticosteroids [ 189, 190]. ] 

Comment: Both “eczematous dermatitis” and “morbiliform maculopapular rash” are pleonasms. Please rephrase
Authors response: Thank you for your comment. We rephrased the sentence
See lines:   376-381 [A morbilliform, also known as maculopapular drug reruption that develops via a common drug hypersensitivity reaction. It is characterized by a symmetrical rash with red macules and papules that may appear one to two weeks after drug exposure. Other symptoms include mottling or plaques, fever, runny nose, and bleachable rash areas [191-193]. As the mucous membranes are typically spared, it is possible to distinguish between mucous membranes less and more erupted [191, 192]. The appearance of a morbilliform (maculopapular) rash on the trunk and extremities was noted in tumor patients following treatment with anti-CTLA-4 therapy [188-203].  .  ]

Comment: “Other symptoms include mottling or plaques, fever, runny nose, and bleachable rash areas  [190-192]. As the mucous membranes are typically spared, it is possible to distinguish between mucous membranes less and more erupted [190, 191].” (lines 409-411) The mention of a “runny nose” should be removed as the next line states mucosae are spared. Please explain the meaning of “mucous membranes less and more erupted”.
Authors response: Thank you very much for your comment. Accordingly, we removed the runny nose and modified the next sentence.
See lines: 378-381 ; the following sentence [As the mucous membranes are typically spared, it is possible to distinguish between mucous membranes less and more erupted ] is replaced by the following one [As the mucous membranes are typically spared, it is possible to distinguish between mucous membranes less and more erupted [191, 192]. The appearance of a morbilliform (maculopapular) rash on the trunk and extremities was noted in tumor patients following treatment with anti-CTLA-4 therapy [188-203].]
Comment: “The appearance of a morbilliform  (maculopapular) rash on the trunk and extremities was noted in tumor patients following treatment with anti-CTLA-4 therapy [189-191].” (lines 411-412) The maculopapular rash is also encountered in patients receiving anti-PD1 or anti-PDL1 treatment.
Authors response: Thank you very much for your comment. We corrected the sentences and involved also both anti-PD-1 and PD-L1
See lines :406-407;  the following sentence [The appearance of a morbilliform (maculopapular) rash on the trunk and extremities was noted in tumor patients following treatment with anti-CTLA-4 therapy [189-191].  ] has been replaced by the following one [The appearance of a morbilliform (maculopapular) rash on the trunk and extremities was noted in tumor patients following treatment with either  anti-CTLA-4, PD-1 or PD-L1 therapy [192, 193] ].
Comment: “Of note, most rashes caused by immunotherapy are associated with a morbilliform reaction [189-191].” (line 413) Please provide details regarding the frequency of this type of reaction.
Authors response: Thank you very much for your comment. We modified the sentences and provided more details
See Lines:383-390; the following sentence [Of note, most rashes caused by immunotherapy are associated with a morbilliform reaction [189-191.   ] are modified and replaced by the following text [As is known, morbilliform rashes are very common in patients treated with anti-CTLA4 therapy. Up to 14 to 26 % of patients receiving ipilimumab and up to 55 % of patients receiving combination anti-CTLA4/PD-1 inhibition therapy  were found to develop morbilliform rashes [188, 191-194 ].. While the lower frequency rates were noted in patients  received  anti-PD-1 and anti-PD-L1 therapies, up to 20 % of  treated patients developed morbilliform rashes  [195. While the highest  morbilliform reactions were found to occurring in less than 2%  of patients received  monotherapy and less than 5% of patients received combination regimens [195 ]... However, early diagnosis and treatment is necessary to continue oncological therapy at an effective dose. [64, 196 ]
Comment: “Eruptions are typically mild and resolve spontaneously two to three months after treatment with anti-CTLA-4 and/or anti-PD-1 inhibitors. [189, 415 192]. “ Please refer to the course of the skin rash in patients continuing ICIs.
Authors response: Thank you very much for your comment. As required,  we referred he text to skin rash
See lines:389-390  The following text [Eruptions are typically mild and resolve spontaneously two to three months after treatment with anti-CTLA-4 and/or anti-PD-1 inhibitors. [189, 192].]  has been replaced by the following one [However, early diagnosis and treatment is necessary to continue oncological therapy at an effective dose. [64, 196 ]  ] 
Comment: “Newer ICIs targeting PD-L1 are associated with less frequent cutaneous adverse events than  those targeting either CTLA-4 or PD-1 receptors [63, 65].” (line 416) Please provide epidemiologic data.
Authors response: Thank you very much for your comment. Accordingly, we provided the epidemiologic data 
See Lines: 391-405….; the following text [ Newer ICIs targeting PD-L1 are associated with less frequent cutaneous adverse events than those targeting either CTLA-4 or PD-1 receptors [63, 65].[ has been replaced by the following one [ Although the incidence of cutaneous adverse effects  range from 30 to 60% in patients treated with ICIs  [196-209 ], the frequency of cutaneous adverse is variable among patients and ICIs-dependent [62 ].  For example, patients received Anti-CTLA-4 monotherapy demonstrated higher incidence of cutaneous adverse effects (44-59%) when compared with those received anti-PD-1 (34-42%) and anti-PD-L1 (up to 20%) monotherapy [62 ] .  whereas the highest incidence of cutaneous adverse effects (5-72%) was noted in patients treated with the combination of  anti-PD-1 and anti-CTLA-4 agents  [196, 198, 199]. Whereas, the severity analysis of cutaneous adverse effects were observed in about 25% of patients treated with anti-CTLA-4 agents [196, 198, 199.and only 2.4% of treated patients have a grade 3 and 4  sever adverse effects [ 200],. More important, the  incidence of cutaneous adverse effects of grades 3 and 4 were found to be much higher in patients treated with anti-PD-L1 monotherapy (7.2%) compared to those observed in patients treated with anti-PD-1 monotherapy (2.3%) or  hose in patients treated with anti-CTLA-4 monotherapy (4.7%)  [ 201 ] 
 While the highest incidence was noted in patients treated anti-PD-L1 in combination with anti-CTLA-4 (14.5%), when compared the incidence of cutaneous adverse effects in patients treated with  anti-PD-1 combined with anti-CTLA-4 therapies (5.4%) [201 ]. ] 
Comment: “The appearance of morbilliform maculopapular rash is attributed to heterogeneous  mechanisms which regulate measles-like rashes following immunotherapy.” (line 418) Please present this mechanisms in detail.
Authors response: Thank you very much for your comment. Accordingly, we described the mechanism in detail
See lines:371-377; the following text [ The appearance of morbilliform maculopapular rash is attributed to heterogeneous mechanisms which regulate measles-like rashes following immunotherapy. Identified rashes in immunotherapy treated cancer patients appear either as itchy, bleaching, erythematous patches or papules on most of the body surface, especially on the head, palms and the soles of the feet  ] has been replaced by the following [Identified rashes in immunotherapy treated cancer patients appear either as itchy, bleaching, erythematous patches or papules on most of the body surface, especially on the head, palms and the soles of the feet [191-193]. The appearance of morbilliform maculopapular rash is mediated by a mechism similar to those of type IV hypersensivity [188-203]. In these mechanisms, the cytotoxic T cells act as effector cells [62, 201, 202]. Thus, during immunotherapy, activated cytotoxic T lymphocytes can directly damage target cells through the release of cytotoxic cytokines, including perforin, granulysin, either granzyme or granzyme B, and through physical interaction via the FasL/FasR signaling pathway. [201-203].  ]
References
Comment: “Identified rashes in immunotherapy treated cancer patients appear either as itchy, bleaching, erythematous patches or papules on most of the body surface, especially on the head, palms and the soles of the feet [191, 421 192].” (lines 419-420) This phrase should be removed as it is repetitive
Authors response: Thank you for your comment. As required, we remove the paragraph.
Comment: “Thus, patients should be carefully examined for blistering, mucous membrane inflammation, epidermal detachment, high fever, and/or swollen lymph nodes several days to one week after the onset of such rashes.” These are sign of Stevens Johnson syndrome or toxic epidermal necrolysis, which are not mentioned. They do not progress from maculopapular rashes, but should be considered in the differential diagnosis.
Authors response: Thank you very much . we modified the text to be more suitable
See lines: 399-402; the following text [Thus, patients should be carefully examined for blistering, mucous membrane inflammation, epidermal detachment, high fever, and/or swollen lymph nodes several days to one week after the onset of such rashes. Topical steroids, topical calcineurin inhibitors, systemic antihistamines, and topical emollients are often used to treat ICI-associated morbilliform maculopapular rash [193-195] A morbilliform rash often starts on the chest and spread to arms, legs, and neck and characterized by the appearance of flat pink or red spots on the skin after the onset of such rashes. Topical steroids, topical calcineurin inhibitors, systemic antihistamines, and topical emollients are often used to treat ICI-associated morbilliform maculopapular rash [202- 204].  .    ] has been modified and replaced by the following one [   ]
Comment: The management of patients developing maculopapular rash during ICI treatment should be stratified according to the severity of this side effect.
Authorse’response: Thank you very much for your comment. We added more information as required
See Line:404-410;  The followeing paragraph [ Treatment of morbilliform maculopapular rash depends on the severity of the rash and is based on the Common Terminological Criteria for Adverse Events (CTCAE) [184] For example, treatment of grade 1 includes symptomatic measures such as application of topical moisturizers and topical potent or high-potency corticosteroids to the affected areas [207- 209]. Whereas, in grade 2, oral antihistamines are also administered to the treatment [210 212]. Systemic corticosteroids (0.5–1 mg/kg/day of prednisone equivalent), In addition to the symptomatic measures used in grades 1–2, they are maintained in grade 3 rashes and only after excluding other diseases requiring specific treatment (e.g. psoriasis). Oral steroids are tapered after improvement of MIT over 4 weeks [210, 212, 213] ] has been added 
.
Authors response:
Comment: “Lichen planus often appears as purple, itchy, flat bumps that develop over several weeks.” (line 429) Please provide a more accurate description of lichen planus skin and mucosal lesions, using academic terms.
Authors response: Thank you very much for your comment. As required we added a new description for lichen planus
Text
See lines:412-414]; the following text [Lichen planus often appears as purple, itchy, flat bumps that develop over several weeks ]  has been replaced by the following text [Lichen planus is a clinically and histopathologically typical, relatively common pruritic papulosquamous skin disease characterized by purple, flattened, polygonal papules that preferentially affect the distal parts of the extremities and the lower back [62, 184].  In addition to the rashes, it can also affect mucous membranes and nails or cause scarring alopecia [214, 215]. 

Comment: “The onset of rashes can occur either immediately or after the start of treatment, especially in patients undergoing  treatment with PD-1 inhibitors [192, 193].” (lines 440-441) Please provide more precise data on the possible duration between the start of ICI therapy and the appearance of lichen planus, on the frequency of this side effect, on the influence of preexistent lichen planus and the mechanisms underlying this association.
Authors response: Thank you very much for your comment. We made the required modification 
Text
See lines:424-425; the following text [The onset of rashes can occur either immediately or after the start of treatment, especially in patients undergoing  treatment with PD-1 inhibitors [192, 193] ]  has been replaced by the following one [In addition to be more frequently than maculopapular rash,  the occurrence of anti-PD-1 and PD-L1 therapy-associated  lichenoid eruption is delayed [193, 202]..  ] 
Comment: “Psoriasis is a skin disorder that causes skin irritation and the formation of scaly, red plaque- like psoriasis [186, 212].” (line 453) This description is incomprehensive.
Authors response: Thank you very much for your comment. As required, we  modified the sentences to more suitable.
Text
See lines:435-436, the following sentences [Psoriasis is a skin disorder that causes skin irritation and the formation of scaly, red plaque- like psoriasis[186, 212]     ]  have been replaced by the following one [Psoriasis is a chronic disease that is mediated by the cells and molecules of both the innate and adaptive immune systems  is mediated by the overactivation of immune system [186, 212]  ]
Comment: “The development of psoriasis in cancer patients is one of the common cutaneous side effects observed for patients following ICI treatment [214, 215].” (lines 455-457) This is inaccurate. Please provide epidemiologic data and the mechanisms underlying this association.
Authors response: Thank you very much for your comment. As required, we provided addition epidemiological data.
Text
See lines 445:483; we added the following paragraph [The development of psoriasis in cancer patients is one of the common cutaneous side effects observed for patients following ICI treatment [224, 241]. ICIs-induced psoriasis can be divided into two types: One of them referred to as a new- onset psoriasis and the other one is the reactivated psoriasis [62, 226].. Data from the European Network on Cutaneous Adverse Events of Oncology Medicines showed that the epidemiological analysis of 115 cases of psoriasis derived from cancer patients treated with ICIs revealed that 30% of the patients  reactivated psoriasis and 70% of the patients exhibited new-onset psoriasis.[]. Of note, patients included in the above mentioned study  86% of patients received anti-PD-1 therapy while only 14% of the patients received anti-PD-L1therapy [227].
The treatment of cancer patients with anti- PD-1 agents is expected to trigger the activation of both innate and adaptive immune systems. Consequently, the cells that may be affected by PD-1 inhibitors such as dendritic cells, T helper cells, and Treg cells are characterized by their ability to secrete cytokines including interferon-gamma, IL-1, IL-17, and IL-22 [228,229].  and thereby play an important role in the pathogenesis of both new-onset psoriasis and reactivated psoriasis [ 229, 246], 
As is known, neutrophils is a key driver of the innate immune response that plays an important role in the axis of psoriasis pathogenesis [228]. Since the migration of neutrophils from blood vessels into the epidermis to form the Munro or Kogoj microabscess is the typical pathological feature of psoriasis [231, 248] . However, the mechanisms whereby, the neutrophils contribute to the occurrence of psoriasis is attributed to their ability to s secrete various cytokines and chemokines such as TNF-α, IL-17, and IL-36 family factors, as well as neutrophil extracellular traps [233 ].  The mechanisms whereby the dendritic cells contribute to the development of psoriasis during the treatment of cancer patients with PD-1 inhibitors is based on the ability of tumor cells and Tumor-Infiltrating Immune Cells ( e.g. macrophages and myeloid dendritic cells ) to express PD-L1 [229, 234. Consequently, PD-1/PD-L1 signal in tumor microenvironment can mediate the inhibition of T-cells [139, 252. Thus, the treatment of cancer patients with PD-1 inhibitors results in the have been demonstrated to directly induce Interferon γ (IFN-γ) production of activated T-cells, release of IL-12  by intra-tumor dendritic cells subpopulations [236, 237] . [. As is widely reported, IL-12 is involved in the regulation of anti-tumor T-cell immunity and stimulation of T-cell proliferation that in turn can secret various cytokines to initiate a positive feedback inflammatory loop [238, 239]. Also, IL-12 has the potential to inhibit eomesodermin,  a key regulator of T-cell exhaustion [240 ].  . To that end, IL-12 is suggested to be  one of the factors that play an important role in the onset and development of psoriasis [241].. Also, the involvement of macrophages in the development of psoriasis is considered. Since the increased macrophages in psoriatic lesions has been reported [242]. Macrophages can also promote psoriatic angiogenesis via mechanisms mediated by the expression of Vascular Endothelial Growth Factor (VEGF), TGF-β, Platelet-Derived Growth Factor (PDGF), and TNF-α [247, 248]. Also, macrophages have been reported to play a role in positive feedback of inflammatory factors including  IL-1 and prokineticin 2 in psoriasis [249 
Cancer patients with a history of psoriasis are more likely to develop cutaneous psoriasis after ICI treatment than inexperienced patients with psoriasis. [250, 251]. ICI- associated psoriasis is related to plaque psoriasis [197, 253]. ICIs-induced psoriasis includes psoriasis vulgaris, lichenoid features, spongiosis, and eosinophil infiltration [66, 254 In many cases lichenoid reactions disappear spontaneously after ICIs discontinuation [146, 189]. Psoriasiform eruptions can be resistant to conventional psoriasis treatments and may require more targeted therapies [66, 69. In some cases, medication with TNF-α inhibitors has successfully treated ICI-induced psoriasis [61, 66]. Cancer patients with pre-existing psoriasis have shown improvement during ICI therapy, suggesting a complex interaction between the immune system and these medications [197, 253] . Common therapeutic modalities for psoriasis include topical steroids, topical vitamin D analogs, topical steroids and topical vitamin D analogs as well as narrowband ultraviolet B light phototherapy [254, 255].  . In addition to skin-directed therapies, systemic treatments include acitretin, methotrexate, and apremilast [254, 256] to the section of psoriasis 
Comment: “Cancer patients with pre-existing psoriasis have shown improvement during ICI therapy, suggesting a complex interaction between the immune system and these medications [218, 219].” The contrary has also been observed. 
Authors response:
Comment: The frequency of bullous disorders induced by ICI treatment, the influence of this treatment on preexistent bullous diseases and the mechanisms underlying ICI-induced bullous disorders should be thoroughly presented.
Authors response:Thank you very much for your comment. As required, re-edited the the section of bullous disorders and included more information about the frequency of the disease and the mechanisms.
Text
See lines:485-515 ; the following paragraph [6.5. Bullous disorders
Bullous diseases are a group of rare autoimmune skin diseases which cause blistering of the skin and/or mucous membranes. The blisters can range from mild to severe and are often filled with fluid [257-260].  ..
Although the exact cause of bullous disease is unknown, it is believed to be caused by autoantibodies directed against the epidermis or the dermoepidermal junction [224, 254] [
Unlike many cutaneous adverse effects that occur within a few months of ICIs treatment, the occurrence of Bullous disorder can appear first several months after the treatment is initiated [261, 262].
Compared to classic bullous disease, bullous disease in patients treated with ICIs has a prolonged pruritic prodromal phase and usually requires treatment with high-dose corticosteroids for effective symptom management [259, 260] [.  . Treatment of ICI-associated bullous disease can be controlled via administration of topical or systemic steroids [149, 2631].. The most common steroid-based therapies used in the treatment of ICI-associated bullous disease include doxycycline, niacinamide, omalizumab, rituximab, dapsone, methotrexate, and plasma exchange [232, 233].
Although bullous disorders are not common as an adverse effect among ICI treated cancer patients [264 ], bullous pemphigoid, the most common blistering disorder, occurs in about 1% of patients treated with PD-1 or PD-L1 inhibitors [34, 195, 264]. Notably, the occurrence of bullous pemphigoid was observed in cancer patients treated sequentially with PD-1 and CTLA-4, but not in those treated with ipilimumab monotherapy [254, 283 ]. In ICI-induced bullous pemphigoid, a longer period of rash-free pruritus and a longer time interval between symptom onset and diagnosis are observed compared to idiopathic bullous pemphigoid. [261, 266].
The induction of  bullous disorders by PD-1/PD-L1 inhibitors is thought to be mediated by both T cell and B cell dysregulation-dependent mechanisms. In T cell-independent humoral immunity, PD-1/PD-L1 blockades can enhance B cell activation, releasing the production of disease-specific autoantibodies [264-267.  Also,. Then, the autoantibodies trigger cross-reactive immunogenicity against basement membrane proteins BP180 and BP230 to  induce bullous pemphigoid [264, 266, 267].  Also,  In T cell-dependent humoral immunity, PD-1 acts as an activator for B cells and enables their interaction with either T helper cells or Treg cells within the germinal centers [268, 269]. T helper cells play a role in the selection and survival of B cells and enable their differentiation into memory B cells or high-affinity antibody-producing plasma cells. While, Treg cells maintain immune balance by suppressing both T helper cells and B cells [270, 271 ]... Inhibition of PD-1 may reduce the suppressive ability of Treg cells and lead to potentially mutated B cells selected by T helper cells [266, 272]. . As a result, abnormal production of low-affinity plasma cells occurs, which contribute to the development of numerous antibody-mediated autoimmune diseases, including bullous pemphigoid. [293, 274 ] .. . In addition to cross-reactivity, the production of autoantibodies against different epitopes, the so-called epitope spreading phenomenon, has been observed in anti-PD-1/PD-L1-associated bullous pemphigoid. [266, 275] [..   
6.6. Vitiligo 
Vitiligo is a chronic, acquired dyschromia that promotes autoimmune aggression against melanocytes, ultimately resulting in the formation of hypochromic or achromic patches and dots on the skin and mucous membranes [13, 260] ..The pathogenesis of vitiligo or vitiligo-like depigmentation is multifactorial and includes individual genetic susceptibility, melanocyte auto-aggression, and failure of immune tolerance mechanisms [55, 58, 59] . The development of vitiligo occurs most frequently in melanoma patients after treatment with ICIs and is less common in patients treated for other cancers [66, 67].. Vitiligo development occurs in melanoma patients receiving anti-CTLA-4 and/or anti-PD-1 inhibitors as mono- or combination therapies. Several studies have indicated that patients receiving combinational therapy with CTLA-4, and PD-1 inhibitors are more likely to develop vitiligo compared to those receiving monotherapy [66, 67].
The development of vitiligo during or after ICIs treatment is clinically significant, as the appearance of vitiligo is not only an indication of immune system activation, but also has positive prognostic implications in melanoma patients [276, 277] ... The paradoxical nature of the association between vitiligo and improved outcome suggests the possibility that depigmentation is attributed to a strong antitumor immune response [278,279.. Conversely, another aspect to be considered is the timing of vitiligo onset in relation to ICI therapy. While some studies report the development of vitiligo-like depigmentation just a few weeks after the treatment has begun, other studies report a delayed onset of the disease by several months after starting therapy [280, 281].. These large differences make it necessary to ensure continuous monitoring of patients undergoing ICIs. To date there is no definitive treatment for ICI-associated vitiligo. Ongoing photoprotection with clothing and sunscreen is the best option to avoid sunburns in these patients [224, 282].. While there is no specific medication to stop ICI-induced vitiligo, there are some drugs that can slow pigmentation loss [283]. The new medications for vitiligo Ruxolitinib (Opzelura) is the most promising class of drug that inhibits both JAK 1 and 2 leading to the treatment facial vitiligo showing a significant improvement [174, 251, 284].
. Although there is increasing evidence suggesting a possible link between immune-related adverse effects and clinical benefit in some cancers [285-188]. Consequently, the more development of immune-related adverse effect, the better response of cancer patients to ICI therapy is expected. 
The relationship between cutaneous adverse reactions and treatment efficacy has been the subject of extensive research, particularly in the context of ICIs in various malignancies. Several studies have investigated the relationship between the timing of cutaneous adverse effects onset and treatment efficacy in patients with different tumor types treated with PD-1/PD-L1 inhibitors [285-188] .
The relationship between immune-related adverse events and treatment efficacy in patients with non-small cancer lung cell (NSCLC) has been investigated in several studies [289-291]..
To evaluate the relationship between immune-related adverse effects occurrence and treatment outcomes with atezolizumab [292]:, pooled data from three Phase 3 clinical trials including (IMpower130) [293, IMpower132 [294]. and IMpower150 [295] have been analyzed. The most common immune-related adverse effects were 28% of skin rash (28%), 15% of liver dysfunction (15%) and 12% of thyroid dysfunction (12%) [288]. However, patients with immune-related adverse effects exhibited longer overall survival (OS) when compared with the treated patients, who exhibited no immune-adverse effects [291, 296] 
Also, in melanoma patients the development of immune-related adverse effects were correlated with treatment efficacy [297-299 ].  
The incidence of immune-related adverse effects among nivolumab treated patients was 68.2% and patients with immune-related adverse effects exhibited a marked difference in overall survival, when compared with treated patients, who do not exhibit immune-related adverse effects [300, 301]. However, the most significant association was noted between immune-related adverse effects such as rash, vitiligo and overall survival in patients with metastatic disease  [288, 302].
6.7. Pruritus 

Comment: The possible correlation between cutaneous adverse effects and ICIs efficacy should be more thoroughly discussed.
Authors response: Thank you very much for valuable comment. We discussed the he correlation between cutaneous adverse effects and ICIs efficacy
Text
See lines:   , the following paragraph [. Although there is increasing evidence suggesting a possible link between immune-related adverse effects and clinical benefit in some cancers [285-188]. Consequently, the more development of immune-related adverse effect, the better response of cancer patients to ICI therapy is expected. 
The relationship between cutaneous adverse reactions and treatment efficacy has been the subject of extensive research, particularly in the context of ICIs in various malignancies. Several studies have investigated the relationship between the timing of cutaneous adverse effects onset and treatment efficacy in patients with different tumor types treated with PD-1/PD-L1 inhibitors [285-188] .
The relationship between immune-related adverse events and treatment efficacy in patients with non-small cancer lung cell (NSCLC) has been investigated in several studies [289-291]..
To evaluate the relationship between immune-related adverse effects occurrence and treatment outcomes with atezolizumab [292]:, pooled data from three Phase 3 clinical trials including (IMpower130) [293, IMpower132 [294]. and IMpower150 [295] have been analyzed. The most common immune-related adverse effects were 28% of skin rash (28%), 15% of liver dysfunction (15%) and 12% of thyroid dysfunction (12%) [288]. However, patients with immune-related adverse effects exhibited longer overall survival (OS) when compared with the treated patients, who exhibited no immune-adverse effects [291, 296] 
Also, in melanoma patients the development of immune-related adverse effects were correlated with treatment efficacy [297-299 ].  
The incidence of immune-related adverse effects among nivolumab treated patients was 68.2% and patients with immune-related adverse effects exhibited a marked difference in overall survival, when compared with treated patients, who do not exhibit immune-related adverse effects [300, 301]. However, the most significant association was noted between immune-related adverse effects such as rash, vitiligo and overall survival in patients with metastatic disease  [288, 302].
 ] has been added

References
See lines: 536   ; the following references [  ] have been added to the references section.

Comment: The effect of systemic corticosteroids and immunosuppressive agents used to control the cutaneous side effects, as well as the optimal management of these patients should be addressed
Authors response: Thank you very much for your valuable comment. As requested, we added more information about the application of systemic corticosteroids the optimal management of cutaneous adverse effects-associated with the treatment of cancer patients with ICIs.
Text
See Lines:536-551   ; the following paragraph [. Although there is increasing evidence suggesting a possible link between immune-related adverse effects and clinical benefit in some cancers [285-188]. Consequently, the more development of immune-related adverse effect, the better response of cancer patients to ICI therapy is expected. 
The relationship between cutaneous adverse reactions and treatment efficacy has been the subject of extensive research, particularly in the context of ICIs in various malignancies. Several studies have investigated the relationship between the timing of cutaneous adverse effects onset and treatment efficacy in patients with different tumor types treated with PD-1/PD-L1 inhibitors [285-188] .
The relationship between immune-related adverse events and treatment efficacy in patients with non-small cancer lung cell (NSCLC) has been investigated in several studies [289-291]..
To evaluate the relationship between immune-related adverse effects occurrence and treatment outcomes with atezolizumab [292]:, pooled data from three Phase 3 clinical trials including (IMpower130) [293, IMpower132 [294]. and IMpower150 [295] have been analyzed. The most common immune-related adverse effects were 28% of skin rash (28%), 15% of liver dysfunction (15%) and 12% of thyroid dysfunction (12%) [288]. However, patients with immune-related adverse effects exhibited longer overall survival (OS) when compared with the treated patients, who exhibited no immune-adverse effects [291, 296] 
Also, in melanoma patients the development of immune-related adverse effects were correlated with treatment efficacy [297-299 ].  
The incidence of immune-related adverse effects among nivolumab treated patients was 68.2% and patients with immune-related adverse effects exhibited a marked difference in overall survival, when compared with treated patients, who do not exhibit immune-related adverse effects [300, 301]. However, the most significant association was noted between immune-related adverse effects such as rash, vitiligo and overall survival in patients with metastatic disease  [288, 302].
] has been added.
Comment: English language editing is required.
Authors response: Thank you very much for your comment. We edited also the English language.

Reviewer 2 Report (New Reviewer)

Comments and Suggestions for Authors

Line 22:  I think this is a very reductive statement on the role of immunotherapy in cancer patients. In many tumors, first of all   Melanoma, followed by Merkel cell carcinoma, NSCLC, TNB cancer, kidney cancer, MSI gastrointestinal and endometrial neoplasms.  ICI had dramatically improved long-term survival with respect to chemotherapy and nowadays it is the standard of care for this neoplasm.

Line 43-44: “immunotherapy is an effective treatment strategy for cancers highly infiltrated by immune cells, such as non-small cell lung cancer”, it is partially true, the better results were observed in cancer with a higher mutational tumor burden, as NSLCC in smokers, skin cancers in sun exposed areas, MSI neoplasms, virus related neoplasms.

Line 46:  systematic or systemic therapy?

Line 54: “dose adjustments” are not an allowed option for ICIs, since the dose is fixed, in case of toxicity, according to the CTACE severity, the treatment could be delayed and eventually restarted or discontinued: these are the international Guidelines (ASCO, ESMO, SITC).

Line 63:  Shown

Line 64 the most common side effect is pruritus, with or   without associated cutaneous rash

Line 71 “to be higher

Line 77 their occurrence

Line 93 unclear

Lines 101-102 unclear

Line 117 check and rephrased better, please

Lines 125-128   repeated concept, reference 66 and 67 (CTLA-4 nanobodies) unrelated to the text

Fig 2 unclear could be replaced by fig 3.    CTLA-4 and its homologous C28 play oppositive effects after binding with their ligands CD80/86 on APC; the affinity and avidity of CTLA4 is higher than CD28.  CTLA4 molecule as internalized and expressed on APC surface as consequence of the recognition of the MHC-peptide complex by the TCR and B7 (Cd80/86) by the co-stimulatory receptor CD28.

Reference 111 not pertinent

Line 220 of note

Line 226. The binding of CTLA-4 to B7 is inhibitory and its affinity and avidity is higher than the binding of CD28 to B7, see fig 5 and I would prefer to stop the note to line 244, to avoid unnecessary complication.

 Line 255-256, incorrect statement:  PD-L1:CD80 cis-heterodimerization inhibited both PD-L1:PD-1 and CD80:CTLA-4 interactions through distinct mechanisms (PMID: 31757674 https://doi.org/10.1016/j.immuni.2019.11.003)

Line 304: widely

Line 308: the potential

Line 309: Blockade of immune checkpoint signaling pathways has been proved effective in restoring suppressed antitumoral cytotoxicity.

Lines 317-340   discussed above and redundant

Reference 175 not pertinent

Table 1 all references are incorrect, check them

Lines 507-508   vitiligo is a mirror of the antitumoral efficacy of ICI, and it was observed and reported only   in melanoma patients, I am not aware of the necessity and indication to contrast or treat it.

Line 520: antihistamines are not effective on treating pruritus induced by ICIs

Comments on the Quality of English Language

should be checked

Author Response

Dear editor
 thank you very much for Comments and Suggestions of the reviewer,
We answered the comment of the reviewer as requested. We hope now the manuscript is improved and meets the quality for publication
On behalf of all my coauthors.

Comment: Line 22:  I think this is a very reductive statement on the role of immunotherapy in cancer patients. In many tumors, first of all   Melanoma, followed by Merkel cell carcinoma, NSCLC, TNB cancer, kidney cancer, MSI gastrointestinal and endometrial neoplasms.  ICI had dramatically improved long-term survival with respect to chemotherapy and nowadays it is the standard of care for this neoplasm.
Authors’ response: thank you very much for your valuable comment. As required, we rephrased the sentences
See lines:23-26    ; the following sentences [ Immunotherapy, particularly that based on blocking of checkpoint proteins offers cancer patients ]  have been rephrased as following [ Immunotherapy, particularly that based on blocking of checkpoint proteins in many tumors, including melanoma, Merkel cell carcinoma, non-small cancer lung cell (NSCLC), Triple-negative breast (TNB cancer), renal cancer, gastrointestinal and endometrial neoplasms is an alternative therapeutic to chemotherapy  ]
Comment: Line 43-44: “immunotherapy is an effective treatment strategy for cancers highly infiltrated by immune cells, such as non-small cell lung cancer”, it is partially true, the better results were observed in cancer with a higher mutational tumor burden, as NSLCC in smokers, skin cancers in sun exposed areas, MSI neoplasms, virus related neoplasms.
Authors ‘response: Thank you for your comment. As requested, we modified and changed the text 
See Lines:46-52; the following sentences [ immunotherapy is an effective treatment strategy for cancers highly infiltrated by immune cells, such as non-small cell lung cancer [2]  ]  have been modified and replaced by the following sentences [Immunotherapy is an effective treatment strategy for cancers that are heavily infiltrated by immune cells, such as non-small cell lung cancer (NSLCC), as well as cancers with a higher mutational tumor burden (MTB), skin cancer, microsatellite instability (MSI) neoplasms, and virus-related neoplasms [2,3]

Comment: Line 46:  systematic or systemic therapy?
Authors’response:  Thank you very much for your comment. We corrected it  
See line: 53; “ Systematic therapy” is changed to “systemic therapy”
Comment: Line 54: “dose adjustments” are not an allowed option for ICIs, since the dose is fixed, in case of toxicity, according to the CTACE severity, the treatment could be delayed and eventually restarted or discontinued: these are the international Guidelines (ASCO, ESMO, SITC).
Authors’ response: Thank you very much for your comment. We modified the sentence to be more suitable .
See lines:63-64; the following sentence [These dermatological side effects can have a significant impact on the well-being and quality of life of cancer patients and, in severe cases, may lead to dose adjustments and interruption or termination of cancer treatment   ] has been rephrase as following [These dermatological side effects can have a significant impact on the well-being and quality of life of cancer patients and, in severe cases, may lead to dose modification and interruption or termination of cancer treatment of cancer treatment [27].]. . 
Comment: Line 63:  Shown
Authors’response: Thank you very much for your comment. Accordingly , we corrected the the word.
See line:70; the word “ schon” to corrected word” shown”
Comment: Line 64 the most common side effect is pruritus, with or   without associated cutaneous rash
Authors’response: thank you very much for your comment. As requested, we replaced the Pruritic maculopapular rash with the “pruritus” [See line: 71]
See line:66; the following [ Pruritic maculopapular rash ]  has been replaced by [ pruritus]
Comment: Line 71 “to be higher
Authors ‘response:  thank you for your comment. We corrected the mistake
See lines:86-87; the o be higher”  has been corrected”  to be higher”   
Comment: Line 77 their occurrence
Authors’ response: thank you very much for your comment. We made the correction.
See Line: 90;  now the “heir occurrence” has been corrected to “their occurrence”
Comment : Line 93 unclear
Authors’response: Thank your very much for your comment. As required, we rephrased the sentence to easy understand.
See lines:104-107; the following sentence [Dysregulation of immune system functioning can occur at various interfaces in the immune cascade.] has been changed and replaced by the following sentence [Immune dysregulation is maladaptive change in molecular control of immune system through various processes mediated by ICIs, component in the pathogenesis of autoimmune diseases and cancers-dependent mechanisms..]   ]
Comment: Lines 101-102 unclear
Authors’response:  Thank you very much for your comment. We rephrased the sentences  to be more clear.
See lines: 105-107; the following sentences [The expression of CTLA-4 in activated T- cells and Treg cells has been reported to play a crucial in the regulation of the thera-peutic efficacy of ICIs in the treatment of autoimmune diseases and cancer]

Comment: Line 117 check and rephrased better, please
Authors’response:  Thank you very much for your comment. As required, we rephrased the sentences as requested.
See lines:125-128; the following sentences [To that end, in addition to to cancer treatment, ICIs can negatively affect host normal tissues leading the development of cutaneous adverse reactions including rashes, dry skin, blisters, itching, and vitiligo. ] have been  rephrased as following [ In summary, in addition to being effective drugs, ICIs carry the risk of multiple and potentially serious immune-related adverse effects in multiple organ systems. ICI-associated side effects include the development of skin reactions, including rashes, dry skin, blisters, itching, and vitiligo. ]

Comment: Lines 125-128   repeated concept.
Authors ‘response: Thank you very much for your comment: Accordingly, we rephrased his paragraph to be more suitable.
See lines:135-140 ; the following paragraph [ T and B cells are key components of the adaptive immune system. Through their immune properties and their interactions with other immune cells and cytokines in their environment, they form a complex network to achieve immune tolerance and maintain the body's homeostasis. As is known, the  activation of naïve T- cells is mediated by two signals. The first one is derived from TCR that recognizes a small part of the antigen to ensure the specificity of the response. So that only T- cells that recognize this antigen will be activated. While the second signal that is known as co-stimulatory signal is provided by the co-stimulatory molecule, CD28 [67, 68].] .
Comment:  reference 66 and 67 (CTLA-4 nanobodies) unrelated to the text
Authors ‘response:  Thank you very much for your comment. We replaced the references by more suitable references.
See lines: 783-787  the following references [67. Cantrell, D. Signaling in lymphocyte activation. Cold Spring Harb Perspect Biol 2015, 7 (6). DOI: 10.1101/cshperspect.a018788
68. Meng, X.; Layhadi, J. A.; Keane, S. T.; Cartwright, N. J. K.; Durham, S. R.; Shamji, M. H. Immunological mechanisms of tolerance: Central, peripheral and the role of T and B cells. Asia Pac Allergy 2023, 13 (4), 175-186. DOI: 10.5415/apallergy.0000000000000128. ] have been added to the references section.
Comment: Fig 2 unclear could be replaced by fig 3.    CTLA-4 and its homologous C28 play oppositive effects after binding with their ligands CD80/86 on APC; the affinity and avidity of CTLA4 is higher than CD28.  CTLA4 molecule as internalized and expressed on APC surface as consequence of the recognition of the MHC-peptide complex by the TCR and B7 (Cd80/86) by the co-stimulatory receptor CD28.
Authors ‘response: Thank you very much for your comment. As request, we removed Fig. 2 and replaced it by Fig.3.
See Fig. 2; also Fig. 4 becomes Fig 3; Fig. 5 becomes Fig. 4; Fig. 6 becomes Fig. 5, and Fig. 7 becomes Fig. 6.
Comment: Reference 111 not pertinent
Authors ‘response: Thank you very much for your comment. We replaced the reference 111 by more suitable one
See Lines:899-900; the following reference [[111] Sansom, D. M. CD28, CTLA-4 and their ligands: who does what and to whom? Immunology 2000, 101 (2), 169-177. DOI: 10.1046/j.1365-2567.2000.00121. x] has been replaced by the following reference [ 112. Sharma, P.; Allison, J. P. Immune checkpoint targeting in cancer therapy: toward combination strategies with curative potential. Cell 2015, 161 (2), 205-214. DOI: 10.1016/j.cell.2015.03.030. ].
Comment: Line 220 of note
Authorse ‘response: Thank you very much for your comment. We corrected it
See line: 207; Of noe is replaced by the corrected “ Of note”
Comment: Line 226. The binding of CTLA-4 to B7 is inhibitory and its affinity and avidity is higher than the binding of CD28 to B7, 
Authors ‘response:  Thank you very much of your comment. Accordingly, we replaced the sentences by the suggested one.
See lines:211-212 ; the following sentences [In fact, binding of CTLA-4 to B7 may generate inhibitory signals to counteract the stimulatory signals [42,76,82, 226 92].] is replaced by your suggested sentences [ The binding of CTLA-4 to B7 is inhibitory and its affinity and avidity is higher than the binding of CD28 to B7 to  generate inhibitory signals that can counteract the stimulatory signals]

Comment: see fig 5 and I would prefer to stop the note to line 244, to avoid unnecessary complication.
Authors ‘response: Thank you very much for your comment.  Accordingly, we removed the following sentences [The ability of CTLA-4 to induces indolamine 2, 3- dioxygenase 244 (IDO) from antigen presenting cells (APCs) is a mechanism whereby the CTLA-4 triggers T cell 245 inhibition. ].
Comment: Line 255-256, incorrect statement:  PD-L1:CD80 cis-heterodimerization inhibited both PD-L1:PD-1 and CD80:CTLA-4 interactions through distinct mechanisms (PMID: 31757674 https://doi.org/10.1016/j.immuni.2019.11.003)
Authors’ response: Thank you very much for comment. Accordingly, we rephrased the sentences.
See lines:236-239;  The following sentences [The binding of both CTLA-4 and PD-1with CD80/CD86 has similar negative effects on T- cell activity  ] were replaced by the following sentences []. Both CTLA-4 and PD-1 binding have similar negative effects on T cell activity, however, the downregulation of CTLA-4 and PD-1 receptors-dependent mechanisms by ICIs is different. Since CTLA-4 expression is restricted to T cells, PD-1 expression is more likely to occur on activated T cells, B cells and myeloid cells [31, 64, 94, 122] x
The following reference [ Zhao, Y.; Lee, C. K.; Lin, C. H.; Gassen, R. B.; Xu, X.; Huang, Z.; Xiao, C.; Bonorino, C.; Lu, L. F.; Bui, J. D.; et al. PD-L1:CD80 Cis-Heterodimer Triggers the Co-stimulatory Receptor CD28 While Repressing the Inhibitory PD-1 and CTLA-4 Pathways. Immunity 2019, 51 (6), 1059-1073.e1059. DOI: 10.1016/j.immuni.2019.11.003. ] to the references section [See lines:925-928]
Comment: Line 304: widely
Authors ‘response:  Thank very much for your comment. We corrected “As is wiedly” to be “  As is widely “ [ See line: 280]

Comment: Line 308: the potential
Authors ‘response: Thank you very much for your comment. We corrected the “he potential” to be “ the potential”[See line : 283 ]
Comment: Line 309: Blockade of immune checkpoint signaling pathways has been proved effective in restoring suppressed antitumoral cytotoxicity.
Authors’ response:  Thank you very much for your comment. Accordingly, we made the required correction.
See lines: 285-286; the following sentences [Blockade of immune checkpoint signaling pathways is essential for the enhancement of immune system to induce antitumor immunity  ] have been replaced by the following sentences [Blockade of immune checkpoint signaling pathways has been proved effective in restoring suppressed antitumoral cytotoxicity  ]

Comment: Lines 317-340   discussed above and redundant
Authors ‘response: Thank you very much for your comment. Accordingly, we removed the mentioned paragraph
Authors ‘response: Thank you very much for your comment. Accordingly, we modified this paragraph to avoid redundance .
Comment: Reference 175 not pertinent
Authors ‘response:  Thank you very much for your comment. The mentioned reference is replaced by more suitable reference
See lines: 1073-1075; the following reference [  ] is replaced by the following reference [Arin, M. J.; Engert, A.; Krieg, T.; Hunzelmann, N. Anti-CD20 monoclonal antibody (rituximab) in the treatment of pemphigus. Br J Dermatol 2005, 153 (3), 620-625. DOI: 10.1111/j.1365-2133.2005.06651.x. ] 
Comment: Table 1 all references are incorrect, check them
Authors’ response: Thank you very much for your comment. Accordingly, we checked the references and replaced hem whenever is required replaced the references with a new one. (See references : [170-176]
Comment: Lines 507-508   vitiligo is a mirror of the antitumoral efficacy of ICI, and it was observed and reported only   in melanoma patients, I am not aware of the necessity and indication to contrast or treat it.
Authors’ response: Thank you very much for your comment. Accordingly, we rephrased the sentences
Text
See lines:  536-538; the following sentences [These drugs, which include corticosteroids, topical Janus kinase inhibitors e.g. ruxolitinib and calcineurin inhibitors, enhance melanocyte growth and bring color back to the skin  ]  have been replaced by the following the sentences [The new medications for vitiligo Ruxolitinib (Opzelura) is the most promising class of drug that inhibits both JAK 1 and 2 leading to the treatment facial vitiligo showing a significant improvement [174, 251, 284].  

Comment: Line 520: antihistamines are not effective on treating pruritus induced by ICIs
Authors’ response: Thank you very much for comment. 

Authors ‘response:567-569; the following sentence [ Milder cases may respond to topical emollients or antihistamines [255-257]] has been replaced by the following  sentence [For grade 1/2 pruritus, moderate to high potency topical corticosteroids, oral antihistamines, and topical emollients are recommended; immunotherapy can usually be continued [209-214]. 
Comments on the Quality of English Language
Authors ‘response:  hank you very much for your comment. Accordingly, we improved the English language

Round 2

Reviewer 1 Report (New Reviewer)

Comments and Suggestions for Authors

Dear authors, 

Thank you for addressing all the concerns raised and for making the appropriate changes. The manuscript has been considerably improved. 

Best regards!

Author Response

Thank you very much for your comment 

This manuscript is a resubmission of an earlier submission. The following is a list of the peer review reports and author responses from that submission.

Round 1

Reviewer 1 Report

Comments and Suggestions for Authors

1. Correct minor grammar mistakes, such as "inhibitors of checkpoint inhibitors" (abstract). some subtitles are not actually nouns ("Lichinoid", "psoriasisform")

2. Paper is "built backwards": first you describe specific cutaneous disorders, then you explain the immune system relevant to CKIs, then you describe the immune mechanisms of CKI cutaneous toxicities. Should be the other way around. 

3. Table 1 is too detailed and, as a result, unreadable. 

Comments on the Quality of English Language

See comments to authors 

Author Response

Author's Reply to the Review Report (Reviewer 1)

Comment1: Correct minor grammar mistakes, such as "inhibitors of checkpoint inhibitors" (abstract). some subtitles are not actually nouns ("Lichinoid", "psoriasisform").

Authors ‘response: Thank you very much for your valuable comment. Accordingly, we corrected the grammar mistakes

See lines:4-6; the sentence [inhibitors of checkpoint inhibitors] has been replaced by the following sentence [on blocking of checkpoint proteins to offer cancer patients an alternative to chemotherapy. Immune checkpoint inhibitors (ICIs)-based therapies have]

See line: 372; the word [lichenoid] is replaced by the word [Lichen planus]

See lines : 373-374; the following sentence [Lichen planus often appears as purple, itchy, flat bumps that develop over several weeks. In the mouth and genital mucosa.  In addition, it can form lacy white patches, sometimes with painful sores] has been added

See lines: 374: the word [Psoriasiform] is replaced by the word [psoriasis], and the sentence [Psoriasiform dermatitis causes] is replaced by the following sentence [Psoriasis is a skin disorder that causes]

Comment 2. Paper is "built backwards": first you describe specific cutaneous disorders, then you explain the immune system relevant to CKIs, then you describe the immune mechanisms of CKI cutaneous toxicities. Should be the other way around. 

Authors ‘response:  Thank you very much for your comment and suggestion. We rearranged and organized the manuscript as required, and added the following text [6. Immunotherapy- Associated Cutaneous Adverse Events

See lines: 298-306; he folloein paragraph  [ ICI-associated Cutaneous adverse reactions are characterized by their delayed occurrence and a longer duration, when compared to the characteristics of the classic chemotherapy- ssociated adverse events [178, 179]. However, depending on the type of cutaneous adverse effects, the time of their occurrence ranges from a few weeks to several months after the treatment has been initiated [180, 181]. Of note, the relationship between cutaneous adverse effects and the dose or time of ICI exposure is not fully understood. ICIs can trigger a variety of skin reactions, which may either represent a reactivation or worsening of an existing dermatosis or a new development [59, 178]. Although the classification of ICI-induced cutaneous adverse effects is still unclear, the most known cutaneous adverse effects mainly from clinical observation and include rash or inflammatory dermatitis that encompass erythema multiforme, lichenoid, eczematous, psoriasiform, morbilliform, and palmoplantar erythrodysesthesia.] was added to the section of  section of cutanous adverse effects

Reference Section

See lines: 603-604 and Lines. 910-923; The following references [[59] Watanabe, T.; Yamaguchi, Y. Cutaneous manifestations associated with ICIs. Front Immunol 202314, 1071983. DOI: 10.3389/fimmu.2023.1071983.

[178]Tomsitz, D.; Ruf, T.; Zierold, S.; French, L. E.; Heinzerling, L. Steroid-Refractory Immune-Related Adverse Events Induced by Checkpoint Inhibitors. Cancers (Basel) 2023, 15 (9). DOI: 10.3390/cancers1509253

[179] Griffiths, C.; de Bruin-Weller, M.; Deleuran, M.; Fargnoli, M. C.; Staumont-Sallé, D.; Hong, C. H.; Sánchez-Carazo, J.; Foley, P.; Seo, S. J.; Msihid, J.; et al. Dupilumab in Adults with Moderate-to-Severe Atopic Dermatitis and Prior Use of Systemic Non-Steroidal Immunosuppressants: Analysis of Four Phase 3 Trials. Dermatol Ther (Heidelb) 2021, 11 (4), 1357-1372. DOI: 10.1007/s13555-021-00558-0.

[180] Blauvelt, A.; Teixeira, H. D.; Simpson, E. L.; Costanzo, A.; De Bruin-Weller, M.; Barbarot, S.; Prajapati, V. H.; Lio, P.; Hu, X.; Wu, T.; et al. Efficacy and Safety of Upadacitinib vs Dupilumab in Adults With Moderate-to-Severe Atopic Dermatitis: A Randomized Clinical Trial. JAMA Dermatol 2021, 157 (9), 1047-1055. DOI: 10.1001/jamadermatol.2021.3023.

[181] Barrios, D. M.; Phillips, G. S.; Geisler, A. N.; Trelles, S. R.; Markova, A.; Noor, S. J.; Quigley, E. A.; Haliasos, H. C.; Moy, A. P.; Schram, A. M.; et al. IgE blockade with omalizumab reduces pruritus related to immune checkpoint inhibitors and anti-HER2 therapies. Ann Oncol 202132 (6), 736-745. DOI: 10.1016/j.annonc.2021.02.016.]

Comment 3: Table 1 is too detailed and, as a result, unreadable. 

Authors ‘response:  Thank you very much for your comment and suggestion. We have reduced the table to be more readable.

See table: Tab. 1

Reviewer 2 Report

Comments and Suggestions for Authors

The reviewed article provides a thorough literature review of cutaneous adverse effects related to immunotherapy. The publication addresses many important issues, primarily focusing on the mechanisms underlying these adverse effects. It takes into account various aspects, including pathophysiological features divided into specific disease states, the mechanistic role of immune checkpoints in cancer and normal tissues, and the molecular basis of these adverse reactions. Notably, there is a well-organized table listing agents used to treat ICIs-induced cutaneous adverse effects. Overall, in my opinion, the article presents a comprehensive review of the discussed topics. However, I suggest that the introduction should include more information regarding the frequency and clinical characteristics of cutaneous adverse effects for individual drugs. Additionally, given the extensive discussion of cutaneous adverse effects, it would be beneficial also to describe those with a non-immunological mechanism, such as phototoxicity.

Author Response

Comment: The reviewed article provides a thorough literature review of cutaneous adverse effects related to immunotherapy. The publication addresses many important issues, primarily focusing on the mechanisms underlying these adverse effects. It considers various aspects, including pathophysiological features divided into specific disease states, the mechanistic role of immune checkpoints in cancer and normal tissues, and the molecular basis of these adverse reactions. Notably, there is a well-organized table listing agents used to treat ICIs-induced cutaneous adverse effects. Overall, in my opinion, the article presents a comprehensive review of the discussed topics. However, I suggest that the introduction should include more information regarding the frequency and clinical characteristics of cutaneous adverse effects for individual drugs. Additionally, given the extensive discussion of cutaneous adverse effects, it would be beneficial also to describe those with a non-immunological mechanism, such as phototoxicity.

Authors’ response: Thanks very much for valuable comment. Accordingly, we have added more information regarding the frequency and clinical characteristics of cutaneous adverse effects for individuals.

See lines:34-50; The following text [These dermatological side effects can have a significant impact on the well-being and quality of life of cancer patients and, in severe cases, may lead to dose adjustments and interruption or termination of cancer treatment [ 27 ICIs-induced cutaneous adverse effects are mediated by immune, non-immune and genetic factors-dependent mechanism The immune-mediated factors (human leukocyte antigen (HLA) allele, genetic polymorphisms) are variable among populations, drug, and phenotype-dependent [27]; non-immune mediated factors, conversely, include abnormalities in genes that encode drug metabolism enzymes, differences in disease type, and drug-related reactions [28].

Although the development of cutaneous adverse effects is common to immunotherapy, their occurrence seems to be time drug type dependent. Pruritic maculopapular rash is one of the most common cutaneous side effects observed in cancer patients treated with both PD-1/PD-L1 and CTLA-4 inhibitors [25, 29,30]. While the development of the neutrophilic dermatoses including pustular eruptions, bullous lupus erythematosus, and pyoderma gangrenosum in the treatment of cancer patients with anti CTLA-4, the ipilimumab [ 31,32].

Although the development of autoimmune bullous diseases in association with immunotherapy are rarely less to occur, the risk of a bullous eruption seems to be higher to occur in cancer patients following the treatment of cancer patients with anti-PD-1 or anti-PD-L1 rather than anti-CTLA-4 [33,34 ] The occurrence of pruritus in cancer patients after the treatment with anti-PD-1 is higher than its occurrence in cancer patients treated with anti-CTLA-4 [35,36 ]. . While the combination of anti-PD-1 (nivolumab) and anti-CTLA-4 (ipilimumab) was found to increase the occurrence of pruritus in cancer patients [37, 38 ],  the incidence of severe cases is low, particularly with anti-PD-L1 treatment [39, 40 ] ] was added to the introduction section

References

The following references

See lines: 530-572; the following references [[27] Sibaud, V. Anticancer treatments and photosensitivity. J Eur Acad Dermatol Venereol 202236 Suppl 6 (Suppl 6), 51-58. DOI: 10.1111/jdv.18200.]

[28 ] Waldman, A. D.; Fritz, J. M.; Lenardo, M. J. A guide to cancer immunotherapy: from T cell basic science to clinical practice. Nat Rev Immunol 202020 (11), 651-668. DOI: 10.1038/s41577-020-0306-5..

[29] Gomes, N.; Sibaud, V.; Azevedo, F.; Magina, S. [Cutaneous Toxicity of Immune Checkpoint Inhibitors: A Narrative Review]. Acta Med Port 2020, 33 (5), 335-343. DOI: 10.20344/amp.12424;

[30] Brahmer, J. R.; Lacchetti, C.; Schneider, B. J.; Atkins, M. B.; Brassil, K. J.; Caterino, J. M.; Chau, I.; Ernstoff, M. S.; Gardner, J. M.; Ginex, P.; et al. Management of Immune-Related Adverse Events in Patients Treated With Immune Checkpoint Inhibitor Therapy: American Society of Clinical Oncology Clinical Practice Guideline. J Clin Oncol 201836 (17), 1714-1768. DOI: 10.1200/JCO.2017.77.6385.].

[ 31] Lopez, A. T.; Khanna, T.; Antonov, N.; Audrey-Bayan, C.; Geskin, L. A review of bullous pemphigoid associated with PD-1 and PD-L1 inhibitors. Int J Dermatol 201857 (6), 664-669. DOI: 10.1111/ijd.13984.

[32 ] Apalla, Z.; Lallas, A.; Delli, F.; Lazaridou, E.; Papalampou, S.; Apostolidou, S.; Gerochristou, M.; Rigopoulos, D.; Stratigos, A.; Nikolaou, V. Management of immune checkpoint inhibitor-induced bullous pemphigoid. J Am Acad Dermatol 202184 (2), 540-543. DOI: 10.1016/j.jaad.2020.05.045.

[33 ] Pintova, S.; Sidhu, H.; Friedlander, P. A.; Holcombe, R. F. Sweet's syndrome in a patient with metastatic melanoma after ipilimumab therapy. Melanoma Res 201323 (6), 498-501. DOI: 10.1097/CMR.0000000000000017.

[ 34] Siegel, J.; Totonchy, M.; Damsky, W.; Berk-Krauss, J.; Castiglione, F.; Sznol, M.; Petrylak, D. P.; Fischbach, N.; Goldberg, S. B.; Decker, R. H.; et al. Bullous disorders associated with anti-PD-1 and anti-PD-L1 therapy: A retrospective analysis evaluating the clinical and histopathologic features, frequency, and impact on cancer therapy. J Am Acad Dermatol 201879 (6), 1081-1088. DOI: 10.1016/j.jaad.2018.07.008. ].

[35 ] Hassel, J. C.; Heinzerling, L.; Aberle, J.; Bähr, O.; Eigentler, T. K.; Grimm, M. O.; Grünwald, V.; Leipe, J.; Reinmuth, N.; Tietze, J. K.; et al. Combined immune checkpoint blockade (anti-PD-1/anti-CTLA-4): Evaluation and management of adverse drug reactions. Cancer Treat Rev 201757, 36-49. DOI: 10.1016/j.ctrv.2017.05.003;

[36 ]. Belum, V. R.; Benhuri, B.; Postow, M. A.; Hellmann, M. D.; Lesokhin, A. M.; Segal, N. H.; Motzer, R. J.; Wu, S.; Busam, K. J.; Wolchok, J. D.; et al. Characterisation and management of dermatologic adverse events to agents targeting the PD-1 receptor. Eur J Cancer 201660, 12-25. DOI: 10.1016/j.ejca.2016.02.010].

[37 ] Larkin, J.; Chiarion-Sileni, V.; Gonzalez, R.; Grob, J. J.; Cowey, C. L.; Lao, C. D.; Schadendorf, D.; Dummer, R.; Smylie, M.; Rutkowski, P.; et al. Combined Nivolumab and Ipilimumab or Monotherapy in Untreated Melanoma. N Engl J Med 2015373 (1), 23-34. DOI: 10.1056/NEJMoa1504030;

[38 ] Robert, C.; Schachter, J.; Long, G. V.; Arance, A.; Grob, J. J.; Mortier, L.; Daud, A.; Carlino, M. S.; McNeil, C.; Lotem, M.; et al. Pembrolizumab versus Ipilimumab in Advanced Melanoma. N Engl J Med 2015372 (26), 2521-2532. DOI: 10.1056/NEJMoa1503093.

[ 39] Postow, M. A.; Chesney, J.; Pavlick, A. C.; Robert, C.; Grossmann, K.; McDermott, D.; Linette, G. P.; Meyer, N.; Giguere, J. K.; Agarwala, S. S.; et al. Nivolumab and ipilimumab versus ipilimumab in untreated melanoma. N Engl J Med 2015372 (21), 2006-2017. DOI: 10.1056/NEJMoa1414428.

[40] Balar, A. V.; Galsky, M. D.; Rosenberg, J. E.; Powles, T.; Petrylak, D. P.; Bellmunt, J.; Loriot, Y.; Necchi, A.; Hoffman-Censits, J.; Perez-Gracia, J. L.; et al. Atezolizumab as first-line treatment in cisplatin-ineligible patients with locally advanced and metastatic urothelial carcinoma: a single-arm, multicentre, phase 2 trial. Lancet 2017389 (10064), 67-76. DOI: 10.1016/S0140-6736(16)32455-] has been added to reference section

Reviewer 3 Report

Comments and Suggestions for Authors

The title of this review is misleading since only figure 7 and one paragraph are dealing with mechanisms of occurence. The rest of the manuscript covers mechanims of anti-PD-1 and anti-CTLA inhibition and cutaneous AEs and their therapeutic treatment options.

Overall, the manuscript is not well structured and a red threat is missing. There are a lot of redundancies across the chapters giving the impression that each chapter has been written by a different author.

Additionally, there are several issues, which should be adressed:

·         I would not name PDAC in connection with an effective treatment strategy (line 47).

·         Endocrine AEs should be added to ICI treatment related AEs (lines 49-51).

·         Cutaneous AEs include immune- and non-immune-mediated and genetic factors (line 53). What do the authors mean?

·         Figure 1 B: PD-1 is currently binding to MHC and the TCR to PD-1L. This should be corrected.

·         The sentence in line 94 is not understandable and the reference (41) is not correct.

·         The sentence (lines 143-144) is misplaced at the current position.

·         Nanobodies in line 300 should read antibodies.

·         Figure legend Figure 3. Line 304. It should read presented instead of represented. There is no CTLA-4: B7 mediated co-stimulation.

·         Figure 5 B. Antagonistic CTLA-4 mAb and agonistic CD28 mAB mixed. The authors should specify this in the figure (legend).  Furthermore, the role of CD28 mAB is nowhere mentioned in the manuscript. The figure legend is not understandable „stimulates T-cells on the surface of APC,…).

·         Authors should check for spacing errors throughout the manuscript (e.g. line 267, line 272).

Author Response

Author's Reply to the Review Report (Reviewer 3)

Please provide a point-by-point response to the reviewer’s comments and either enter it in the box below or upload it as a Word/PDF file. Please write down "Please see the attachment." in the box if you only upload an attachment. A template can be found here.

Comments and Suggestions for Authors

Comment: The title of this review is misleading since only figure 7 and one paragraph are dealing with mechanisms of occurence. The rest of the manuscript covers mechanims of anti-PD-1 and anti-CTLA inhibition and cutaneous AEs and their therapeutic treatment options.

Authors ‘response: Thank you very much for your valuable comment. Accordingly we changed he title to be more related the context of the manuscript.

See Lienes:2-3; the following title [Immunotherapy- Associated Cutaneous Adverse Events:  Mechanisms of Occurrence ]  has been replaced by the following tile [ Immune Check Point inhibitors- Associated Cutaneous Adverse Events:  Mechanisms of Occurrence  ]

Comment: Overall, the manuscript is not well structured and a red threat is missing. There are a lot of redundancies across the chapters giving the impression that each chapter has been written by a different author.

Authors ‘response: Thank you very much for your comment. As required, we structured the manuscript  and improved it  to meet your request

See the main text:  The manuscript is newly structured

Comment: I would not name PDAC in connection with an effective treatment strategy (line 47).

Authors ‘response: thank you for your comment, but I do not know, what you mean with “ PDAC in connection with an effective treatment strategy “ since I do not find this word.

  • Comment: Endocrine AEs should be added to ICI treatment related AEs (lines 49-51).

Authors ‘response: thank you very much your valuable comment. Accordingly, we modified the abstract and the introduction to include ICIs-induced endocrine adverse effects.

See lines : 28-30, The following sentence [ The most observed ICIs-induced adverse effects are mediated by the activation of autoreactive T-cells leading to the occurrence of various adverse effects like autoimmune diseases. Gastrointestinal toxicity, endocrine toxicity, and dermatologic toxicity 5, 5-12].]has been added to the introduction section

Abstract

See lines: 4-5; the following sentences [Immunotherapy, particularly that based on blocking of checkpoint proteins to offer cancer patients an alternative to chemotherapy. Immune checkpoint inhibitors (ICIs)-based therapies] have been added to abstract.

Introduction section

See lines:28-30; the following sentences [The most observed ICIs-induced adverse effects are mediated by the activation of autoreactive T-cells leading to the occurrence of various adverse effects like autoimmune diseases. Gastrointestinal toxicity, endocrine toxicity, and dermatologic toxicity  [5-12].] have been added to the introduction.

Comment: Cutaneous AEs include immune- and non-immune-mediated and genetic factors (line 53). What do the authors mean?

Authors ‘response: Thank you very much for your valuable comment. We apologize for the uncorrected formulation of the sentence. We mean that the ICIs-induced cutaneous adverse effects are mediated by immune, non-immune and genetic factors-dependent mechanisms. Accordingly, we modified the text as following:

See lines:36-39; the following sentence [ Cutaneous adverse effects include immune- and non-immune-mediated and genetic factors.] has been replaced by he following sentence [ICIs-induced cutaneous adverse effects are mediated by immune, non-immune and genetic factors-dependent mechanism The immune-mediated factors (human leukocyte antigen (HLA) allele, genetic polymorphisms) are variable among populations, drug, and phenotype-dependent [27]; non-immune mediated factors, conversely, include abnormalities in genes that encode drug metabolism enzymes, differences in disease type, and drug-related reactions [28]. ]

Comment: Figure 1 B: PD-1 is currently binding to MHC and the TCR to PD-1L. This should be corrected.

Authors ‘response: Thank you very much for your variable comment. Accordingly, we corrected the mislabeling.

See Figures: Fig. 1B

Comment: The sentence in line 94 is not understandable and the reference (41) is not correct.

Authors ‘response: Thank you very much for your comment, we apricate that. You mean the following sentence [Cross-reactivity between antigens on target tumor cells and self-antigens on normal host tissues is the primary mechanism through which immunotherapy overcomes tumor resistance [41] ] that is located in the lines: 76-78  ]. We modified the sentence to be easy to understand and we corrected the reference [41]

See lines:79-81; the following sentence [Cross-reactivity between antigens on target tumor cells and self-antigens on normal host tissues is the primary mechanism through which immunotherapy overcomes tumor resistance ] has been modified and replaced by the following sentence[Cross-reactivity of antigens on target tumor cells is the primary mechanism that enables immunotherapy to target tumor cells without to interact with the self-antigens located on host normal tissues [53].]

Reference [ 41] has been corrected is now [53]

See reference section lines: 608-610 [ 53] Yoest, J. M. Clinical features, predictive correlates, and pathophysiology of immune-related adverse events in immune checkpoint inhibitor treatments in cancer: a short review. Immunotargets Ther 2017, 6, 73-82. DOI: 10.2147/ITT.S126227...]

Comment: The sentence (lines 143-144) is misplaced at the current position.

Authors ‘response: Thank you very much for your valuable comment. Accordingly, we corrected the sentence

See lines: 351-352: the following sentence [It presents itchy, purplish papules and plaques that commonly appear on the wrists, lower back, and ankles. 70, [74] has been modified and replaced by the following sentence [Lichen planus often appears as purple, itchy, flat bumps that develop over several weeks. In the mouth and genital mucosa. In addition, it can form lacy white patches, sometimes with painful sores [146, 185, 186]

  • Comment: Nanobodies in line 300 should read antibodies.

Authors ‘response:thank you very much for your comment. You mean the following ford Nanobodies located in line in line: 281

See line: 120; the word nanobadies  is replaced by the antibodies

Comment: Figure legend Figure 3. Line 304. It should read presented instead of represented. There is no CTLA-4: B7 mediated co-stimulation.

Authors ‘response: Thank you very much for your comment. You mean line: 285.  Accordingly, we corrected he mistakes

See lines: 123-124 ; the following sentence [ which are presented by MHC on APCs, and CTLA-4: B7 and CD28:B7-mediated stimulation.] has been corrected

Comment: Figure 5 B. Antagonistic CTLA-4 mAb and agonistic CD28 mAB mixed. The authors should specify this in the figure (legend).  Furthermore, the role of CD28 mAB is nowhere mentioned in the manuscript. The figure legend is not understandable „stimulates T-cells on the surface of APC,…).

Authors ‘response: Thank you very much for valuable comment.  We apologize for the mistakes. According, we corrected the mistakes and modified the legend to be easy to understand.

See lines; 167-169, The following about CD28 [CD28 is the best studied co-stimulatory glycoprotein, whos main function is crucial for the co-stimulation of naive T lymphocytes [106, 111], in addition to protects T cell from apoptosis and to increase cell proliferation and cytokine secretion [ 106, 111].  has been added to  the section of CTLA-4

References

See lines:756-757 and 768-769;    ; The following references

[106] Esensten, J. H.; Helou, Y. A.; Chopra, G.; Weiss, A.; Bluestone, J. A. CD28 Costimulation: From Mechanism to Therapy. Immunity 201644 (5), 973-988. DOI: 10.1016/j.immuni.2016.04.020.

[111] Sansom, D. M. CD28, CTLA-4 and their ligands: who does what and to whom? Immunology 2000101 (2), 169-177. DOI: 10.1046/j.1365-2567.2000.00121.x. . ] have been added to the references section

See lines: 185-190; the following legend [Figure 5. CTLA-4/B7 and CD28/CD80/86 pathway-dependent tumor immune escape. A) The binding of CTLA-4 to B7 and CD28 to CD80/86 on the surface of immune effector cells (T cells) results in the suppression of T cell receptor (s) (TCR) to recognize major histocompatibility (MHC) molecules on the surface of antigen presenting cells (APC), which are activated in response to antigens produced by tumor cells. B) The inhibition of the binding of CTLA-4 to B7 and the binding of CD28 to CD80/86 by monoclonal antibodies for CTLA-4 or CD28 stimulates T- cells on the surface of APC, triggering tumor cell death. ] has been modified  as following [ Figure 5. CTLA-4/B7 and CD28/CD80/86 pathway-dependent tumor immune escape. A) The binding of CTLA-4 to B7 and CD28 to CD80/86 on the surface of immune effector cells (T cells) results in the suppression of T cell receptor (s) (TCR) to recognize major histocompatibility (MHC) molecules on the surface of antigen presenting cells (APC), which are activated in response to antigens produced by tumor cells. B) The inhibition of the binding of CTLA-4 to B7 and the binding of CD28 to CD80/86 by monoclonal antibodies for CTLA-4 (e.g. ipilimumab and tremelimumab ) or CD28 (e.g. (super antagonist anti-CD28 antibodies)   ) stimulates T- cells  to trigger tumor cell death]

Comment: Authors should check for spacing errors throughout the manuscript (e.g. line 267, line 272).

Authors ‘response: Thank you very much for the valuable comment. Accordingly, we checked the whole manuscript the space errors

Round 2

Reviewer 3 Report

Comments and Suggestions for Authors

The authors have tried to improve the manuscript, which was partly successful. However, there are still many spelling errors and hard to understand  sentences within the current version. 

The wrong binding partners in Figure 1 b have not been corrected.

The statement in the legend of Figure 5 that binding of CD28 to CD80/86 results in the suppression of TCRs  to recognize MHC molecules is simply wrong. Firstly,  the interaction of CD28 to CD80/86 leads to T cell stimulation,

Secondly, the TCR does not recognize MHC molecules. It recognizes the antigen bound on the MHC. Binding of CTLA4 to CD80/CD86 inhibits the transmission of the signal of TCR-Antigen binding inside the cell.

In response to the rebuttal letter, I just would like to clarify that PDAC ist the abbreviation for pancreatic ductal adenocarcinoma, and as mentioned previously, i would not name this cancer type as an example for the efficay of ICI.

Author Response

Authors’ response to Reviewer 3

Comment1: The authors have tried to improve the manuscript, which was partly successful. However, there are still many spelling errors and hard to understand sentences within the current version. 
Authors’ response: Thank you very much for your comment. Accordingly, we corrected the manuscript overall and simplified the sentences to be easy to understand

Comment 2: The wrong binding partners in Figure 1 b have not been corrected.

Authors’ response: Thank you very much for your comment. We corrected Fig. 1b
Comment 3. The statement in the legend of Figure 5 that binding of CD28 to CD80/86 results in the suppression of TCRs  to recognize MHC molecules is simply wrong. Firstly,  the interaction of CD28 to CD80/86 leads to T cell stimulation, Secondly, the TCR does not recognize MHC molecules. It recognizes the antigen bound on the MHC. Binding of CTLA4 to CD80/CD86 inhibits the transmission of the signal of TCR-Antigen binding inside the cell
Authors’ response: Thank you very much for your comment. We corrected and modified the legened of the figure 
See lines:203-205; the following sentence [The binding of CTLA-4 to B7 and CD28 to CD80/86 on the surface of immune effector cells (T cells) results in the suppression of T cell receptor (s) (TCR) to recognize major histocompatibility (MHC) molecules on the surface of antigen presenting cells (APC), which are activated in response to antigens produced by tumor cells ] has been replaced by the following one [Both TCR- major histocompatibility (MHC) and CD28-CD80/86 signaling pathways are essential for T-cell activation. Up on co-stimulation of the two signaling pathways, naïve T cells become active and express CTLA-4, which binds with CD80/86 molecules to inactivate T cells.].
Comment 4: In response to the rebuttal letter, I just would like to clarify that PDAC is the abbreviation for pancreatic ductal adenocarcinoma, and as mentioned previously, i would not name this cancer type as an example for the efficay of ICI.
Authors’ response: Thank you very much for comment. Accordingly, we replaced the pancreatic ductal adenocarcinoma by other tumor type as example for the efficacy of ICI as following

Text:
See line: 46; the following tumor type [ pancreatic ductal adenocarcinoma ] by the following [Such as non-small cell lung cancer [2] ] 
References 
See lines: 450-452;  Reference 2 [ Wang, H.; Chen, L.; Qi, L.; Jiang, N.; Zhang, Z.; Guo, H.; Song, T.; Li, J.; Li, H.; Zhang, N.; et al. A Single-Cell Atlas of Tumor-Infiltrating Immune Cells in Pancreatic Ductal Adenocarcinoma. Mol Cell Proteomics 2022, 21 (8), 100258. DOI: 10.1016/j.mcpro.2022.100258.] has been replaced by the following reference [Guo, X.; Zhang, Y.; Zheng, L.; Zheng, C.; Song, J.; Zhang, Q.; Kang, B.; Liu, Z.; Jin, L.; Xing, R.; et al. Global characterization of T cells in non-small-cell lung cancer by single-cell sequencing. Nat Med 2018, 24 (7), 978-985. DOI: 10.1038/s41591-018-0045-3.  ]
